# LADDERS OF THOUGHT: A SELF-EVOLVING CURRICULUM OF PROGRESSIVELY SIMPLIFIED REASONING TRACES

## ABSTRACT

Large language models (LLMs) excel at reasoning when scaled to hundreds of billions of parameters, but **small- and mid-scale** models remain brittle reasoners even with knowledge distillation (KD). We present **Ladders-of-Thought (LoT)**, a framework that improves reasoning by combining progressive question rewrites with a self-evolving curriculum. LoT automatically generates semantically faithful but easier variants of reasoning problems, organizes them into difficulty buckets using step-based measures, and employs a self-evolving bandit scheduler to allocate training adaptively. Evaluated on two reasoning domains, **math** and **multi-hop** reasoning, across 1-8B models from different families, LoT consistently improves over KD. It delivers large gains on arithmetic tasks (e.g., +32 percentage points on AddSub, +25pp on SVAMP), +2–8pp improvements on in-domain test splits, and strong though dataset-dependent benefits on multi-hop reasoning (e.g., +16pp on QASC, +25pp on StrategyQA). LoT also converges faster than staged curricula, highlighting the value of adaptive progression. These results show that progressive rewrites coupled with adaptive curricula provide a simple yet effective recipe for strengthening reasoning in smaller LLMs.

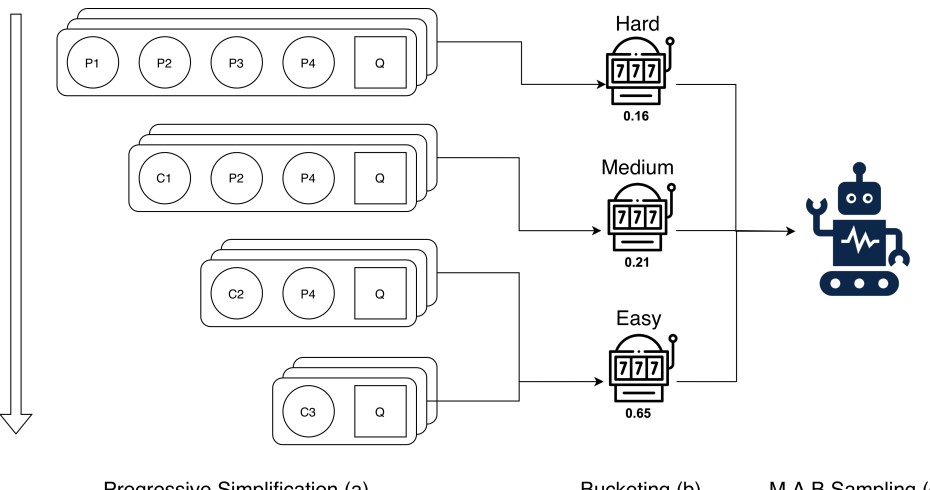

Progressive Simplification (a)  Bucketing (b)  M.A.B Sampling (c)

Figure 1: Overview of our framework. (a) **Progressive simplification**: original reasoning questions are rewritten into semantically faithful but progressively easier variants, forming a difficulty ladder. (b) **Step-based difficulty measure bucketing**: each question is assigned a difficulty score based on the number of required reasoning steps. This score is then used to place each example in a bucket. (c) **Self-evolving curriculum**: a multi-armed bandit scheduler adaptively selects examples from different buckets to maximize student learning progress.

# 1 INTRODUCTION

Large language models (LLMs) have demonstrated remarkable progress on complex reasoning benchmarks, especially when augmented with test-time prompting strategies such as chain-of-thought (CoT) reasoning (Wei et al., 2022; Zhang et al., 2022b), self-consistency (Wang et al., 2022), and structured search methods including tree-of-thoughts (Yao et al., 2023), cumulative reasoning (Zhang et al., 2023), and DUP (Zhong et al., 2024). These approaches highlight the power of explicit reasoning traces in guiding LLMs toward more accurate and robust answers.

> *Our focus is on small- and mid-scale LLMs, where limited capacity, brittle chain evaluation, and large student–teacher gaps make reasoning training especially challenging.*

Despite recent advances, most improvements are concentrated in very large models. Smaller models, while cheaper and more efficient, often fail to benefit from CoT-style prompting and remain brittle reasoners. A key limitation is their poor ability to generalize learned reasoning beyond a specific dataset. This challenge has motivated extensive work on reasoning distillation from large to small models, spanning standard distillation (Hinton et al., 2015; Ho et al., 2022; Magister et al., 2022; Mitra et al., 2023; Fu et al., 2023), symbolic distillation (West et al., 2021), verifier-assisted methods (Li et al., 2023; Zhang et al., 2024; Liu et al., 2023), knowledge-augmented reasoning (Kang et al., 2023), and self-consistent objectives (Wang et al., 2023a). While encouraging, these approaches struggle when the gap between student and teacher is large: small models often overfit to surface heuristics instead of acquiring transferable reasoning skills (Wang et al., 2023b; Li et al., 2025).

Curriculum learning (CL) offers a natural remedy. CL suggests that ordering training examples from easy to hard improves both sample efficiency and generalization (Bengio et al., 2009; Matiisen et al., 2019; Soviany et al., 2022; Narvekar et al., 2020). Adaptive curricula, which dynamically select training examples, often work even better (Jiang et al., 2015; Kong et al., 2021). While CL has been explored in in-context learning (Liu et al., 2024) and reinforcement learning (Chen et al., 2025; Parashar et al., 2025), its potential for supervised fine-tuning of reasoning remains underexplored. A major obstacle is defining difficulty for reasoning problems: length, number of inferential steps, and information structure all interact in complex ways (Jin et al., 2024; Wang et al., 2025; Shi et al., 2025).

**Our Approach.** We introduce **Ladders-of-Thought (LoT)**, a framework for training stronger reasoning in small- and mid-scale LLMs through a combination of *progressive question rewrites* and *self-evolving curricula*. Our method builds on three insights: (1) Reasoning questions can be automatically rewritten into progressively easier versions while preserving semantics, forming a natural "ladder" of difficulty. (2) The minimal number of reasoning steps provides a principled difficulty measure for organizing training buckets. (3) A self-evolving curriculum scheduler, framed as a multi-armed bandit, can adaptively allocate training to difficulty levels where the student learns fastest, avoiding rigid or suboptimal schedules.

**Contributions.** This paper makes three contributions:

- We introduce a progressive rewrite framework that generates semantically faithful but easier variants of reasoning problems, creating a principled difficulty ladder.
- We propose an adaptive, self-evolving curriculum scheduler that dynamically allocates training across difficulty buckets using bandit-based updates.
- We demonstrate through extensive experiments that LoT improves pass@5 accuracy, accelerates convergence, and strengthens out-of-distribution generalization for small- and mid-scale LLMs.

# 2 BACKGROUND

We briefly review the foundations of our approach: chain-of-thought (CoT) distillation, curriculum learning, and multi-armed bandit scheduling.

**Chain-of-Thought Distillation.** CoT distillation transfers reasoning ability from a large teacher to a smaller student by supervising on teacher-generated rationales (Ho et al., 2022; Chae et al., 2023). Given $\mathcal{D} = \{(x^{(i)}, y^{(i)})\}$, we prompt the teacher with zero-shot CoT instructions (Wei et al., 2022; Zhang et al., 2022b) to obtain rationales $r^{(i)}$. Training instances are formatted as

$$\text{Question: } x^{(i)} \quad \text{Answer: } r^{(i)}, y^{(i)}.$$

The student autoregressively generates $r^{(i)}$ and $y^{(i)}$, optimized via negative log-likelihood:

$$\mathcal{L}_{\text{CoT}}(\theta) = -\sum_i \Big( \sum_j \log P_\theta(r_j^{(i)} \mid r_{<j}^{(i)}, x^{(i)}) + \sum_j \log P_\theta(y_j^{(i)} \mid y_{<j}^{(i)}, r^{(i)}, x^{(i)}) \Big).$$

This encourages the student to reproduce step-by-step reasoning and final answers.

**Curriculum Learning.** Curriculum learning (Bengio et al., 2009) presents data in a structured order. A difficulty function $d(x)$ partitions $\mathcal{D}$ into buckets $\{\mathcal{B}_1, \ldots, \mathcal{B}_K\}$, ordered by difficulty. A curriculum defines a sequence of sampling distributions $\{p_t\}_{t=1}^T$, where $p_t(b)$ is the probability of drawing from bucket $\mathcal{B}_b$ at step $t$. Fixed curricula move gradually from easy to hard, while self-evolving ones adjust $p_t$ based on model progress.

**Multi-Armed Bandits.** Self-evolving curricula can be framed as a multi-armed bandit (MAB) problem, where each bucket $\mathcal{B}_k$ corresponds to an arm. At step $t$, the scheduler selects arm $a_t \in \{1, \ldots, K\}$ according to $p_t$, samples from $\mathcal{B}_{a_t}$, and receives reward $r_t$ (e.g., validation improvement). The objective is to minimize regret

$$R_T = \max_k \sum_{t=1}^T r_t^{(k)} - \sum_{t=1}^T r_t,$$

where $r_t^{(k)}$ is the reward had arm $k$ been played. Strategies such as $\epsilon$-greedy and Boltzmann exploration balance exploration with exploitation. We employ such a scheduler to adapt $p_t$ online.

# 3 METHOD: LADDERS-OF-THOUGHT (LOT)

LoT constructs curricula for reasoning tasks through two key components: (i) *progressive rewrites*, which generates graded versions of each question by injecting intermediate reasoning steps (Figure 2), and (ii) *step-based difficulty labeling*, which assigns a consistent measure of problem difficulty. These components together yield difficulty-labeled question sets that can be organized into either staged or adaptive self-evolving curricula (Figure 1).

## 3.1 PROGRESSIVE REWRITES

We start with a question–solution pair $(q, s)$ where the question $q$ contains an explicit set of premises $\mathcal{P} = \{p_1, p_2, \ldots, p_m\}$ and the solution is expressed as a chain-of-thought (CoT) sequence $s = (c_1, c_2, \ldots, c_n)$. Each reasoning step derives a new conclusion $c_i$ from a small set of antecedents $A_i \subseteq \mathcal{P} \cup \{c_1, \ldots, c_{i-1}\}$; for example, $p_1 + p_2 \Rightarrow c_1$ and then $c_1 + p_3 \Rightarrow c_2$.

**Rewrite operation.** Rather than merely appending conclusions to the context, we *replace* the antecedents of each step by the derived conclusion. Concretely, let $q^{(0)} = q$. For $i = 1, \ldots, n$, form

$$q^{(i)} = \big(q^{(i-1)} \setminus A_i\big) \cup \{c_i\}.$$

Intuitively, if $p_1$ and $p_2$ entail $c_1$, we remove $p_1, p_2$ from the question and insert $c_1$ instead, yielding an easier instance. Applying this transformation step-by-step produces a sequence $q^{(0)}, q^{(1)}, \ldots, q^{(n)}$ of strictly decreasing difficulty, terminating when the answer is trivial (or explicitly recoverable) in the context (Figure 2).

**Practical generation.** We prompt a capable instruction-tuned LLM to (i) identify $A_i$ for each CoT step and (ii) produce the simplified question $q^{(i)}$ while preserving semantics and well-posedness. The rewriting model need not coincide with the teacher used for CoT supervision; in practice, we may use a strong CoT generator as the teacher and a separate model for controlled rewriting. This procedure pairs every complex question with progressively easier counterparts, forming the backbone of our curriculum.

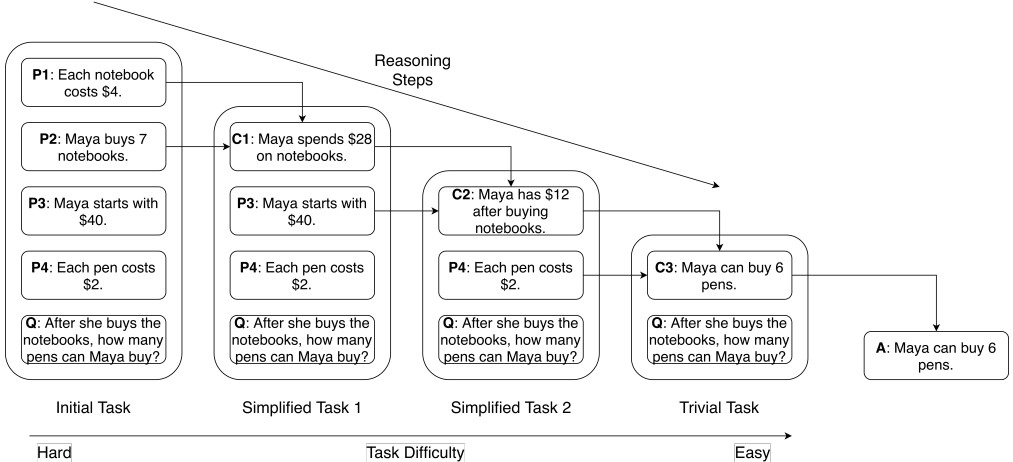

Figure 2: **Progressive rewrites.** Each task can be converted into a series of standalone premises ($P1$, $P2, \dots$). Each reasoning step combines two pieces of information to make a conclusion ($C$), or new piece of information. After each step of reasoning, the total amount of information is smaller, giving an easier sub-question to solve. Thus, progressively easier questions arise naturally from step-by-step problem solving, while preserving semantics and solvability (no answer leakage), since each rewrite replaces a subset of premises with their logically entailed conclusion.

**Comparison to decomposition.** This simplification differs from problem decomposition methods such as Simonds & Yoshiyama (2025), which generate related but distinct subproblems. Our rewrites retain the original problem identity while replacing subsets of premises with intermediate conclusions, i.e., they are the *same* task presented with precomputed inferences in the premise.

## 3.2 DIFFICULTY LABELING VIA STEP DEFINITION

We define the difficulty of a reasoning example by the model-estimated minimal number of steps required to reach a solution, denoted $\phi(x)$. For instance, a math problem that requires three arithmetic operations has $\phi(x) = 3$. Rather than relying on raw chain-of-thought (CoT) length, which can be inflated by verbosity or stylistic padding, our progressive rewriting procedure enforces a one-step decrement at each stage (e.g., $3 \to 2 \to 1 \to 0$). Thus $\phi(x)$ aligns directly with the number of rewrites available for each example, providing a consistent and interpretable difficulty measure. The rewriting model is given explicit instructions on what constitutes a reasoning step to maintain consistent granularity, and we manually spot-check examples to verify the monotonic decrease. This process yields well-calibrated step counts that serve as interpretable difficulty labels. Training data are then bucketed by these labels, $\mathcal{B}_k = \{x : \phi(x) \in I_k\}$, providing a structured progression from easier to harder questions. Complete prompts for step counting and rewriting are provided in Appendix A.8.

## 3.3 CURRICULUM CONSTRUCTION

The difficulty-labeled questions naturally form a curriculum. Because the distribution of step counts is often imbalanced, we group adjacent levels into buckets (e.g., 1–3, 4–5, and 5+ steps as "easy," "medium," and "hard"). These buckets support both staged and self-evolving curriculum strategies.

A simple baseline is the *staged curriculum*, where buckets are ordered by difficulty and the model trains on one bucket at a time for a fixed number of steps. This provides a straightforward schedule against which self-evolving methods can be compared.

For adaptivity, we follow the multi-armed bandit (MAB) framework of Matiisen et al. (2019), which treats each bucket $\mathcal{B}_k \in \{\mathcal{B}_1, \dots, \mathcal{B}_K\}$ as an arm. At step $t$, the learner selects an arm $a_t$, trains on samples from bucket $\mathcal{B}_{a_t}$, and receives a reward derived from validation performance.

The Q-value update is

$$Q_{t+1}(a) = \alpha \, r_t(a) + (1 - \alpha)Q_t(a),$$

with learning rate $\alpha$ and $Q_0(a) = 0$.

Every $m$ steps, we compute rewards as

$$r_t(a) = \text{Acc}_t(a) - \overline{\text{Acc}}_t(a),$$

where $\overline{\text{Acc}}_t(a)$ is an exponential moving average with smoothing coefficient $\beta$. This measures the accuracy gain relative to baseline.

Buckets are then sampled either from a Boltzmann distribution

$$\pi_t(a) \propto \exp(Q_t(a)/\tau),$$

with temperature $\tau$, or via an $\epsilon$-greedy policy that chooses the best bucket with probability $1 - \epsilon$ and explores otherwise.

This bandit-based scheduler dynamically focuses training on the levels that yield the greatest marginal improvement, producing a self-evolving curriculum. The full training procedure, including progressive rewrites, bucketization, and adaptive scheduling, is summarized in Algorithm 2 (see Appendix A.1 for details).

---

**Algorithm 1:** Ladders-of-Thought (LoT): Training with Progressive Rewrites and Self-evolving Curriculum

---

**Input:** Original data $\mathcal{D}_{\text{orig}}$, teacher $T$, rewriting model $R$, student $S_\theta$, budget $S$ steps, buckets $\{\mathcal{B}_k\}$.

**Output:** Fine-tuned student $S_\theta$.

**Progressive Rewriting.** For each $(x, y) \in \mathcal{D}_{\text{orig}}$: Generate rationale $r$ and answer $y$ from $T$;

   Iteratively rewrite $x$ with $R$ into $x^{(d)}$ that is one step easier ($\phi(x^{(d+1)}) = \phi(x^{(d)}) - 1$).

   Collect $(x^{(d)}, r^{(d)}, y^{(d)}, \phi(x^{(d)}))$ until trivial.

**Bucketization.** Group examples by step count $\phi(x)$ into buckets $\{\mathcal{B}_k\}$.

**Training with Bandit Curriculum.** Initialize bandit over buckets. **for** $t = 1$ **to** $S$ **do**

   Sample batch $\mathcal{M}_t$ according to bandit distribution.

   Update $S_\theta$ with CoT loss on $\mathcal{M}_t$: rationale + answer tokens.

   Every $E$ steps (evaluation interval): evaluate on held-out validation splits $\mathcal{B}_k^{\text{val}}$; compute

      rewards; update bandit to adjust sampling probabilities.

**end**

**return** $S_\theta$

---

# 4 EXPERIMENTS

We evaluate whether progressive rewrites combined with a self-evolving curriculum improve reasoning generalization. Our experiments focus on two questions: (i) Does LoT provide consistent gains over strong baselines across models and domains? (ii) How do rewrite depth and curriculum scheduling affect performance?

## 4.1 SETUP

We study two domains: **math** and **multi-hop reasoning**. For math, models are trained on GSM8K Cobbe et al. (2021) and evaluated on its test split plus AddSub, ASDiv, MultiArith, and SVAMP Hosseini et al. (2014); Miao et al. (2020); Roy & Roth (2015); Patel et al. (2021). For multi-hop, models are trained on EntailmentBank Dalvi et al. (2021) and tested on its split plus StrategyQA, OpenBookQA, QASC, and MuSiQue Geva et al. (2021); Mihaylov et al. (2018); Yang et al. (2018); Khot et al. (2020); Trivedi et al. (2022).

We evaluate OPT-1.3B/2.7B Zhang et al. (2022a) and Pythia-1.4B/2.8B Biderman et al. (2023), using knowledge distillation (KD) from a strong CoT teacher. Baselines include: (i) the base model, (ii) CoT KD on original data, and (iii) **LoT (ours)**: KD with rewrites under a self-evolving curriculum.

Performance is reported as pass@5 accuracy [1]. We also report mean $\pm$ standard error and greedy decoding pass@1 accuracies in Appendix A.4.

## 4.2 MAIN RESULTS

Tables 1 and 2 summarize pass@5 accuracy across both domains and model families. LoT consistently outperforms KD on original data, with especially large gains on math reasoning. For example, on OPT-2.7B, AddSub accuracy jumps from 8.26 to 40.37 (+32.11 percentage points), and SVAMP from 19.06 to 44.15 (+25.09 percentage points).

Improvements are also evident on in-domain test splits: GSM8K rises from 31.01 to 33.97 (+2.96), while EntailmentBank improves by +3–8 percentage points across all model families. Across architectures, Pythia-1.4B improves on ASDiv from 21.36 to 40.78 (+19.42), while Pythia-2.8B gains +20.18 on AddSub and +20.74 on SVAMP.

Two trends stand out in math reasoning. First, LoT yields the largest gains on smaller, compositional arithmetic datasets such as AddSub, ASDiv, and SVAMP. These datasets differ substantially from the GSM8K training distribution, highlighting LoT's strength in improving out-of-distribution generalization. Second, while improvements on GSM8K itself are more modest (+2–3 points), LoT consistently prevents degradation and provides robustness, suggesting that introducing easier rewrites does not harm in-domain accuracy while improving transferability.

| Methods | GSM8K | AddSub | ASDiv | MultiArith | SVAMP |
|---------|-------|--------|-------|------------|-------|
| **OPT-1.3B** | | | | | |
| Base | 3.79 | 1.83 | 4.05 | 2.22 | 5.69 |
| CoT KD | 27.75 | 9.17 | 20.23 | 63.33 | 20.74 |
| LoT (Ours) | 31.16 (+3.41) | 33.03 (+23.86) | 41.75 (+21.52) | 68.33 (+5.00) | 38.46 (+17.72) |
| **OPT-2.7B** | | | | | |
| Base | 3.34 | 1.83 | 4.21 | 3.33 | 6.69 |
| CoT KD | 31.01 | 8.26 | 26.38 | 71.11 | 19.06 |
| LoT (Ours) | 33.97 (+2.96) | 40.37 (+32.11) | 45.95 (+19.57) | 80.56 (+9.45) | 44.15 (+25.09) |
| **Pythia-1.4B** | | | | | |
| Base | 2.96 | 0.00 | 4.85 | 1.67 | 8.03 |
| CoT KD | 26.00 | 3.67 | 21.36 | 59.44 | 19.73 |
| LoT (Ours) | 28.35 (+2.35) | 24.77 (+21.10) | 40.78 (+19.42) | 63.33 (+3.89) | 38.46 (+18.73) |
| **Pythia-2.8B** | | | | | |
| Base | 3.71 | 1.83 | 6.63 | 4.44 | 9.70 |
| CoT KD | 33.43 | 11.93 | 31.88 | 74.44 | 24.08 |
| LoT (Ours) | 32.98 (−0.45) | 32.11 (+20.18) | 46.76 (+14.88) | 72.78 (−1.66) | 44.82 (+20.74) |

Table 1: Pass@5 accuracy (%) on GSM8K and out-of-distribution math benchmarks. Each entry shows absolute accuracy with $\Delta$ relative to CoT KD. LoT consistently improves generalization, with the largest gains on AddSub, ASDiv, and SVAMP (+15–30 percentage points). ($\Delta$s are rendered in green/red for increases/decreases.)

For multi-hop reasoning, LoT provides both in-domain and out-of-domain benefits when trained on EntailmentBank. In-domain accuracy rises on the EntailmentBank test split (+3–8), showing that rewrites help the model capture inference patterns more reliably. Out-of-domain, LoT delivers strong improvements on QASC (+4–16) and StrategyQA (+17–25), and also boosts MuSiQue substantially for OPT-1.3B (+25) and Pythia-2.8B (+2.8).

Although LoT improves substantially on QASC and StrategyQA, we observe some regressions on MuSiQue and OpenBookQA. Both tasks lie far outside the supervision domain: models are trained only on EntailmentBank, while MuSiQue and OpenBookQA rely more heavily on factual retrieval, entity grounding, and multi-evidence aggregation than on compositional inference. In these settings, stronger sensitivity to step-structured reasoning patterns and reduced exposure to factual variability in training may limit transfer. These results suggest that LoT provides the largest

---

[1] Pass@5 is computed by drawing 5 samples per query with temperature=0.5 and top-$p = 0.95$, and counting success if any matches the verified answer.

benefits when the target task shares the same underlying inferential structure as the curriculum, and that complementary mechanisms (e.g., retrieval augmentation) may be required to support transfer to knowledge-centric QA.

Overall, LoT delivers improvements across all four model checkpoints and both reasoning domains. Its benefits are **architecture-agnostic** and extend beyond in-domain test sets to multiple out-of-distribution benchmarks, though the magnitude of gains is more uniform in arithmetic reasoning than in multi-hop tasks.

| Methods | EntailmentBank | QASC | OpenBookQA | StrategyQA | MuSiQue |
|---|---|---|---|---|---|
| **OPT-1.3B** | | | | | |
| Base | 22.0 | 17.2 | 14.8 | 32.4 | 2.2 |
| CoT KD | 36.0 | 46.0 | 47.4 | 22.6 | 14.2 |
| LoT (Ours) | 41.0 (+5.0) | 52.8 (+6.8) | 43.6 (–3.8) | 47.8 (+25.2) | 39.2 (+25.0) |
| **OPT-2.7B** | | | | | |
| Base | 21.0 | 28.8 | 12.6 | 33.6 | 2.2 |
| CoT KD | 40.0 | 56.2 | 46.0 | 54.0 | 43.6 |
| LoT (Ours) | 41.0 (+1.0) | 60.4 (+4.2) | 47.8 (+1.8) | 51.2 (–2.8) | 24.0 (–19.6) |
| **Pythia-1.4B** | | | | | |
| Base | 13.0 | 45.0 | 34.2 | 24.4 | 12.0 |
| CoT KD | 36.0 | 55.2 | 44.6 | 36.4 | 42.0 |
| LoT (Ours) | 39.0 (+3.0) | 56.6 (+1.4) | 36.8 (–7.8) | 53.2 (+16.8) | 39.2 (–2.8) |
| **Pythia-2.8B** | | | | | |
| Base | 10.0 | 39.0 | 32.4 | 29.2 | 18.8 |
| CoT KD | 32.0 | 32.2 | 28.2 | 55.6 | 42.2 |
| LoT (Ours) | 40.0 (+8.0) | 48.8 (+16.6) | 37.8 (+9.6) | 60.0 (+4.4) | 45.0 (+2.8) |

Table 2: Pass@5 accuracy (%) on EntailmentBank (in-domain) and four out-of-domain multi-hop benchmarks. LoT improves EntailmentBank by +3–8 percentage points across model families and yields strong gains on QASC (+4–16) and StrategyQA (+17–25). Performance is more mixed on OpenBookQA and MuSiQue (some regressions for smaller models; Pythia-2.8B still improves). Δ values are relative to CoT KD (green/red = increase/decrease).

## 4.3 LARGER STUDENTS: QWEN2.5–7B AND LLAMA3.1–8B

To evaluate whether Ladders-of-Thought scales beyond small and mid-sized students, we additionally trained Qwen2.5–7B and Llama3.1–8B models on the same GSM8K-based LoT curriculum. Results are shown in Table 3.

| Methods | GSM8K | AddSub | ASDiv | MultiArith | SVAMP |
|---|---|---|---|---|---|
| **Qwen2.5-7B** | | | | | |
| Base | 21.61 | 5.50 | 11.17 | 15.00 | 10.70 |
| KD | 73.84 | 50.46 | 77.67 | 98.33 | 60.54 |
| LoT (Ours) | 74.60 (+0.76) | 70.64 (+20.18) | 74.92 (–2.75) | 98.89 (+0.56) | 71.57 (+11.03) |
| **Llama3.1-8B** | | | | | |
| Base | 16.38 | 35.78 | 29.45 | 17.78 | 30.77 |
| KD | 53.22 | 25.69 | 50.32 | 92.22 | 40.13 |
| LoT (Ours) | 56.10 (+2.88) | 56.88 (+31.19) | 47.90 (–2.42) | 93.33 (+1.11) | 50.50 (+10.37) |

Table 3: Pass@5 accuracy (%) on GSM8K and out-of-distribution math benchmarks. Δ indicates absolute change vs KD.

The results show that LoT scales effectively to larger student models. For both Qwen2.5-7B and Llama3.1-8B, LoT matches and sometimes slightly improves over KD on GSM8K and yields substantially larger gains on out-of-distribution tasks. Qwen2.5-7B LoT achieves strong improvements on AddSub (+20.2) and SVAMP (+11.0), while Llama3.1–8B shows similar boosts (+31.2 AddSub, +10.4 SVAMP). These gains mirror the trends observed at the 1-3B scale: LoT

mainly enhances compositional generalization rather than in-distribution accuracy alone. Overall, the results indicate that LoT is not limited to small models and continues to strengthen transfer beyond the training distribution as model size increases.

## 4.4 ABLATION: REWRITE DEPTH

Table 4 shows that rewrite depth has a pronounced effect on performance. Introducing shallow rewrites ($\leq 1$) yields the largest single jump in accuracy (+28.48 percentage points on average), and performance continues to increase up to $\leq 3$, especially on benchmarks requiring multi-step arithmetic composition (e.g., +6.11 on MULTIARITH at $\leq 3$ and +15.05 on SVAMP at $\leq 2$). However, using all rewritten variants leads to diminishing or negative returns, suggesting that excessive exposure to very easy variants can dilute the core reasoning signal and reduce generalization.

To better understand this trend, Appendix A.5 presents a distributional analysis of minimal reasoning steps under different rewrite depths (Table 13 and Figure 5). As rewrite depth increases, the training distribution becomes increasingly skewed toward low-step (i.e., easier) instances. Taken together, these results indicate that LoT benefits from a moderate curriculum ladder: shallow-to-intermediate rewrites broaden exposure to simpler reasoning structures, while preserving a sufficient range of difficulty to avoid over-regularizing the model toward trivial problems.

| Depth | GSM8K | AddSub | ASDiv | MultiArith | SVAMP | Average |
|---|---|---|---|---|---|---|
| 0 | 10.46 | 6.42 | 9.22 | 17.78 | 7.02 | 10.18 |
| $\leq 1$ | 30.55 (+20.09) | 26.61 (+20.19) | 40.61 (+31.39) | 67.78 (+50.00) | 27.76 (+20.74) | 38.66 (+28.48) |
| $\leq 2$ | 28.81 (−1.74) | 32.11 (+5.50) | 48.22 (+7.61) | 71.11 (+3.33) | 42.81 (+15.05) | 44.61 (+5.95) |
| $\leq 3$ | 32.07 (+3.26) | 30.28 (−1.83) | 47.73 (−0.49) | 77.22 (+6.11) | 43.14 (+0.33) | 46.09 (+1.48) |
| All | 31.16 (−0.91) | 33.03 (+2.75) | 41.75 (−5.98) | 68.33 (−8.89) | 38.46 (−4.68) | 42.55 (−3.54) |

Table 4: Pass@5 accuracy (%) when varying maximum rewrite depth. Performance improves sharply when adding shallow rewrites ($\leq 1$), continues to grow up to depth 3, and declines when all rewrites are included. Deltas are relative to the row above (green = improvement, red = decrease).

## 4.5 ABLATION: CURRICULUM SCHEDULING

We compare four curriculum strategies: (i) *Flat Sampling* (random training without curriculum), (ii) *Staged Curriculum (Easy→Hard)*, (iii) *Staged Curriculum (Hard→Easy)*, and (iv) *Self-evolving Curriculum (ours)*.

Figure 3 and Figure 4 highlight the importance of curriculum design. LoT's Self-evolving Curriculum achieves both the fastest convergence and the highest final accuracy, outperforming all fixed schedules. Easy→Hard also improves over Flat sampling, confirming that sequencing problems from simple to complex is more effective than random order. By contrast, Hard→Easy performs worst across the board, lagging in both early and late training. This supports the intuition that exposing models to difficult problems before they have acquired simpler reasoning patterns hinders progress.

Interestingly, Flat sampling often shows reasonable early learning speed, but plateaus at lower accuracy. LoT combines the best of both worlds: it retains early learning efficiency while ultimately achieving stronger final performance. This indicates that adaptivity, rather than a fixed progression, is key for balancing efficiency and generalization.

## 4.6 OVERHEAD OF LOT

LoT adds two sources of overhead: offline rewrite generation and the online MAB scheduler. Rewrite generation is performed once before training, and its token counts and cost estimates are reported in Appendix A.2. During training, we profiled the wall-clock time across all models and found that the MAB scheduler accounts for only 3-8% of the total runtime. The remaining compute is identical to standard supervised fine-tuning. Thus, LoT introduces minimal computational overhead in practice.

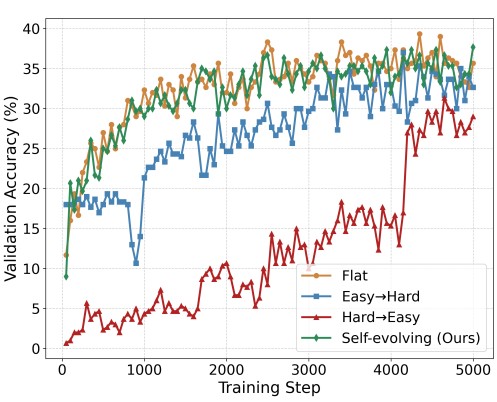 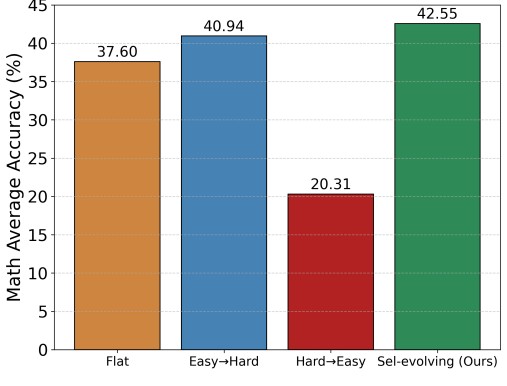

Figure 3: GSM8K validation accuracy over training steps under different curriculum strategies. **Self-evolving (Ours)** (green) and **Flat** (orange) achieve both faster learning and high final accuracy. **Easy→Hard** (blue) is moderately effective, while **Hard→Easy** (red) consistently underperforms.

Figure 4: Average test accuracy across math reasoning benchmarks under different curriculum strategies. **Self-evolving (Ours)** (green) achieves the best performance. Both **Easy→Hard** (blue) and Self-evolving outperform **Flat** (orange), showing that introducing easier problems first leads to stronger learning, while **Hard→Easy** (red) harms performance.

### 4.7 DISCUSSION

Taken together, these analyses show that: (1) LoT consistently boosts reasoning performance, with especially large gains on OOD arithmetic benchmarks; (2) Rewrite depth should be moderate—shallow to intermediate levels provide strong generalization benefits, while excessive depth can hurt; and (3) Curriculum scheduling strongly affects outcomes, with self-evolving strategies clearly outperforming static or reversed schedules, underscoring the importance of curriculum direction and adaptivity.

Overall, these findings suggest that LoT provides a principled recipe for enhancing reasoning models: use faithful but easier rewrites, structure them into a moderate-depth ladder, and adaptively adjust exposure to maximize sample efficiency and generalization. Importantly, LoT achieves these gains with minimal computational overhead, since rewrite generation is performed entirely offline and the MAB scheduler adds only a small fraction of total training time.

## 5 RELATED WORKS

**LLM Reasoning and Distillation.** To transfer reasoning ability to compact LLMs, many works explore distillation (Xu et al., 2024; Yang et al., 2024). Supervised fine-tuning on teacher-generated CoT traces improves small models (Mitra et al., 2023; Magister et al., 2022; Ho et al., 2022; Gu et al., 2023), with variants such as symbolic distillation (West et al., 2021), verifier-assisted training (Liu et al., 2023; Zhang et al., 2024), knowledge-augmented objectives (Kang et al., 2023), and white-box supervision using hidden states (Deng et al., 2023). Despite progress, distillation often breaks down when the student–teacher gap is large, leading to overfitting to shallow heuristics and poor generalization (Li et al., 2025).

**Question Decomposition.** A line of recent work improves reasoning by decomposing questions into smaller sub-tasks or auxiliary queries. Least-to-Most Prompting (Zhou et al., 2022) and Self-Ask (Press et al., 2023) generates a sequence of sub-questions at inference time, without modifying the underlying training distribution. Divide-or-Conquer (Wu et al., 2024) similarly constructs sub-questions but focuses on disentangling decomposition from solving to study which component is easier to distill. LADDER (Simonds & Yoshiyama, 2025) produces hierarchical supervision but still introduces additional sub-problems rather than modifying the original instance. In contrast, LoT does not decompose problems into multiple auxiliary tasks. Instead, it performs progressive

simplification of the same question, replacing antecedent premises with their entailed intermediate conclusions. This preserves semantic equivalence while inducing a structured, monotonic difficulty ladder tied directly to minimal reasoning depth.

**Rationale Refinement and Process-Supervision.** LoT is also related to methods that refine rationales or generate structured intermediate representations. Self-Refine (Madaan et al., 2023) and Reflexion (Shinn et al., 2023) iteratively improve model outputs via self-feedback loops, while Program-of-Thoughts (Chen et al., 2022) and PAL (Gao et al., 2023) disentangle computation from reasoning using executable programs. These approaches operate on generated rationales or outputs, rather than rewriting the input problem itself, and do not yield difficulty-aligned variants of training examples. Moreover, unlike fixed curricula or staged supervision used in some earlier reasoning pipelines, LoT pairs its automatically simplified variants with a non-stationary multi-armed-bandit scheduler, yielding an adaptive training curriculum that evolves with student performance.

**Curriculum Learning.** Curriculum learning (CL) suggests ordering examples from easy to hard to accelerate training and improve generalization (Bengio et al., 2009; Narvekar et al., 2020; Soviany et al., 2022). Extensions include self-paced (Jiang et al., 2015) and adaptive methods (Matiisen et al., 2019; Kong et al., 2021). For LLMs, curricula have been studied in in-context learning (Liu et al., 2024) and reinforcement learning (Shi et al., 2025; Chen et al., 2025; Parashar et al., 2025). Closest to our setting, Chen et al. (2025) also propose self-evolving curricula, but in RL optimization rather than supervised fine-tuning.

**Difficulty Estimation.** Difficulty measures are critical to CL. Prior work has used proxy signals such as MCTS heuristics (Wang et al., 2025), dataset-provided difficulty labels (Chen et al., 2025), or model hit rates (Shi et al., 2025). Other analyses show that longer chains help only when they add true inferential depth (Jin et al., 2024). We instead introduce a step-based measure grounded in the minimal number of reasoning steps, which directly aligns with our progressive rewrites and avoids noisy proxies such as raw CoT length.

## 6 CONCLUSION

We introduced **Ladders-of-Thought (LoT)**, a framework that combines progressive rewrites with an adaptive self-evolving curriculum to improve reasoning in small- to mid-scale LLMs. Our experiments on math and multi-hop reasoning demonstrate that LoT consistently outperforms strong knowledge distillation and curriculum baselines, delivering substantial gains in out-of-distribution arithmetic tasks (e.g., +32 percentage points on AddSub, +25pp on SVAMP), modest but robust improvements on in-domain test sets (GSM8K, EntailmentBank), and dataset-dependent benefits on multi-hop reasoning (notably +25pp on StrategyQA). LoT also accelerates convergence compared to flat or staged curricula, highlighting the value of adaptivity in balancing efficiency with final performance. These findings show that carefully structured training signals—semantically faithful rewrites organized into adaptive curricula—provide a principled recipe for strengthening reasoning in smaller LLMs without requiring more scale or data. We believe LoT offers a practical foundation for future reasoning-focused training pipelines and can complement other emerging curriculum-based strategies.

## LIMITATIONS

Our study focuses on small- to mid-scale LLMs (1–8B parameters); scalability to larger foundation models remains untested. LoT also depends on a capable generator for progressive rewrites—low-quality or unfaithful rewrites may add noise, and the balance between fidelity and diversity is not fully explored. Evaluation is limited to English math and text-only multi-hop benchmarks; extending to multilingual, multimodal, and interactive domains (e.g., vision–language or embodied agents) is a natural next step. Finally, LoT's mixed results on certain multi-hop tasks indicate that benefits are dataset-dependent, raising open questions about which reasoning settings gain most from progressive curricula.

## REPRODUCIBILITY STATEMENT

We have made every effort to ensure the reproducibility of our results. All datasets used in this work are publicly available. We provide details of data preprocessing, rewrite generation, and filtering rules in Appendix A.3. Model architectures (OPT and Pythia) are open-source, and all training hyperparameters, curriculum schedules, and evaluation settings are fully specified in Section 4 and Appendix A.3. We will release our training scripts, curriculum scheduler implementation, and rewrite datasets to facilitate replication and extension by the community.

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

# A  APPENDIX

## A.1  ALGORITHMS

Algorithm 2 presents the Self-Evolving Curriculum Scheduler, a non-stationary multi-armed bandit (MAB) strategy that dynamically allocates training batches across curriculum buckets of increasing difficulty. The scheduler maintains Q-values for each bucket, reflecting recent improvements in model accuracy, and uses these values to guide bucket selection according to either a Boltzmann exploration policy or an $\epsilon$-greedy policy. Periodically, the algorithm evaluates the model on a balanced validation set, computes the reward as the gain over a running accuracy baseline, and updates both the Q-values (via temporal difference learning) and the baselines (via exponential moving average). This design allows the scheduler to adaptively focus training on buckets that yield the greatest learning progress while still preserving exploration.

Algorithm 3 defines the auxiliary procedure VALIDATEBALANCED, which ensures fair assessment of performance across curriculum buckets. The method constructs a validation set that samples an equal number of items from each bucket, evaluates the model independently on each subset, and returns per-bucket accuracies. These balanced evaluations are used by the scheduler (Algorithm 2) to compute bucket-wise rewards and update the learning signals that drive curriculum adaptation.

---

**Algorithm 2:** Self-Evolving Curriculum Scheduler (Non-stationary MAB)

---

**Require:** Buckets $\mathcal{C} = \{c_1, \ldots, c_N\}$ (ordered by difficulty); learning rate $\alpha \in (0, 1]$; EMA
        coefficient $\beta \in (0, 1]$; validation period $m$ (steps); policy
        `policy` $\in \{$`boltzmann`, `epsilon_greedy`$\}$; temperature $\tau > 0$ (Boltzmann);
        exploration rate $\epsilon \in [0, 1]$ ($\epsilon$-greedy)
**Initialize** Q-values $Q_0(c) \leftarrow 0$ and accuracy baselines $\overline{\mathrm{Acc}}_0(c) \leftarrow 0$ for all $c \in \mathcal{C}$
**Initialize** step counter $t \leftarrow 0$
**while** *training not converged* **do**
    $t \leftarrow t + 1$
    `// --- Select a bucket (action) ---`
    **if** `policy` $=$ `boltzmann` **then**
        $\pi_t(c) \propto \exp\big(Q_{t-1}(c)/\tau\big)$                        `// normalize over` $c \in \mathcal{C}$
        Sample $c_t \sim \pi_t(\cdot)$
    **else if** `policy` $=$ `epsilon_greedy` **then**
        With prob. $1 - \epsilon$: $c_t \leftarrow \arg\max_{c \in \mathcal{C}} Q_{t-1}(c)$; else sample $c_t$ uniformly from $\mathcal{C}$
    **end**
    **TrainStep** on a mini-batch from bucket $c_t$   `// one or more gradient updates`
    `// --- Periodic validation and updates ---`
    **if** $t \bmod m = 0$ **then**
        $\{\mathrm{Acc}_t(c)\}_{c \in \mathcal{C}} \leftarrow$ `ValidateBalanced`$(\mathcal{C})$
        **foreach** $c \in \mathcal{C}$ **do**
            $r_t(c) \leftarrow \mathrm{Acc}_t(c) - \overline{\mathrm{Acc}}_{t-1}(c)$         `// improvement over running`
            `baseline`
            $Q_t(c) \leftarrow \alpha \cdot r_t(c) + (1 - \alpha) \cdot Q_{t-1}(c)$     `// TD(0) on non-stationary`
            `reward`
            $\overline{\mathrm{Acc}}_t(c) \leftarrow (1 - \beta) \cdot \overline{\mathrm{Acc}}_{t-1}(c) + \beta \cdot \mathrm{Acc}_t(c)$         `// EMA baseline`
        **end**
    **else**
        **foreach** $c \in \mathcal{C}$ **do**
            $Q_t(c) \leftarrow Q_{t-1}(c);$   $\overline{\mathrm{Acc}}_t(c) \leftarrow \overline{\mathrm{Acc}}_{t-1}(c)$
        **end**
    **end**
**end**

---

---

**Algorithm 3:** VALIDATEBALANCED (helper)

---

**Require:** Buckets $\mathcal{C}$; validation sampler that draws an equal number of items per bucket
Build validation set $\mathcal{V} = \bigcup_{c \in \mathcal{C}} \mathcal{V}(c)$ with $|\mathcal{V}(c)|$ equal across buckets
**foreach** $c \in \mathcal{C}$ **do**
   |    Evaluate current model on $\mathcal{V}(c)$ to obtain accuracy $\text{Acc}_t(c)$
**end**
**return** $\{\text{Acc}_t(c)\}_{c \in \mathcal{C}}$

---

## A.2 DATASET STATISTICS

For all experiments, we use OpenAI GPT-5-mini as both the rewriter and teacher model to generate the progressive rewrite curricula. Rewrite generation is performed entirely offline. Table 5 reports the corresponding input/output token counts and cost estimates.

| Dataset | # Train Qs | Input Tokens | Output Tokens | GPT-5-Mini Batch API Cost[2] |
|---------|-----------|--------------|---------------|------------------------------|
| GSM8K | 7,473 | 5,200,915 | 14,550,446 | $15.20 |
| EntailmentBank | 1,836 | 2,127,678 | 6,518,870 | $6.78 |

Table 5: Rewrite-generation overhead. All rewrites are generated once offline. Costs estimated using the GPT-5-mini batch API.

To ensure that rewritten questions preserve semantic fidelity and form a coherent difficulty ladder, we conducted a 500-sample rewrite quality audit evaluated by GPT-5. Each rewrite was assessed along three criteria:

- **Question validity**: whether the rewritten question is clear, solvable, and self-contained.
- **Difficulty decrease**: whether the rewrite is strictly easier than the original.
- **Answer preservation**: whether solving the rewritten question yields the same answer.

Across the 500 sampled rewrites, we find that:

- **99.2%** were valid and solvable,
- **98.0%** exhibited a clear decrease in difficulty, and
- **98.6%** preserved the original answer.

These results confirm that progressive rewrites reliably maintain semantic identity while producing well-controlled difficulty reductions.

Table 6 provides step-count distributions for all dataset splits used in training, rewriting, validation, and testing.

| Dataset | Split | 0 | 1 | 2 | 3 | 4 | 5 | 6–7 | 8–15 |
|---------|-------|---|---|---|---|---|---|-----|------|
| GSM8K | Train (all) | 7320 | 7352 | 6920 | 5063 | 2989 | 1601 | 1100 | 577 |
| | Validation | 100 | 96 | 87 | 90 | 92 | 90 | 74 | 10 |
| | Test | - | - | - | - | - | - | - | - |
| EntailmentBank | Train (all) | 1660 | 1642 | 1226 | 745 | 421 | 258 | 268 | 134 |
| | Validation | 100 | 99 | 94 | 96 | 95 | 59 | 46 | 16 |
| | Test | 1 | 30 | 29 | 14 | 12 | 8 | 3 | 3 |

Table 6: Step-count distributions for all splits of GSM8K and EntailmentBank. GSM8K test data lacks step annotations (shown as "–").

---

[2]Pricing as of September 2025.

### A.3 EXPERIMENTAL SETUP DETAILS

#### A.3.1 ENVIRONMENT DETAILS

All experiments were conducted on cluster nodes equipped with 4 **NVIDIA RTX A6000 GPUs** (48GB VRAM each), 4 CPU cores, and 64GB of host memory.

**Training.** Models were fine-tuned using `PyTorch`, with the `trl` and `accelerate` libraries handling supervised fine-tuning and multi-GPU execution.

**Evaluation.** Performance was assessed using the `lm-eval-harness`, with our multi-stage answer verification pipeline integrated into the evaluation loop (see Section A.3.4.

#### A.3.2 HYPERPARAMETERS

We list below the key hyperparameters fed to the HuggingFace TRL trainer. Unless otherwise noted, all other hyperparameters follow library defaults.

```
"max_steps": 5000 (Math) / 1000 (Multi-hop),
"per_device_train_batch_size": 8,
"gradient_accumulation_steps": 1,
"max_length": 2048,
"logging_steps": 1,
"learning_rate": 1e-5,
"weight_decay": 0.05,
"warmup_ratio": 0.1,
"lr_scheduler_type": "constant",
```

Validation is performed 100 times during training. The interval is set to 50 steps for math reasoning and 10 steps for multi-hop reasoning. In validation the generation arguments are set to:

#### A.3.3 BUCKETING BY STEP COUNT

To reduce variance and maintain balanced sampling, we group questions by their reasoning *step count* into buckets. Specifically, questions with shorter derivations (0, 1, 2, or 3 steps) are each assigned their own bucket, while questions requiring four or more steps are merged into a single "4+" bucket. This grouping scheme has two advantages: (i) it preserves granularity for very short reasoning chains, which differ substantially in difficulty, and (ii) it avoids fragmentation of the long-tail distribution of high-step examples, which are sparse and uneven across datasets. All curriculum schedules and sampling strategies described in the main text are applied over these step-count buckets.

#### A.3.4 ANSWER VERIFICATION PROCEDURE

Evaluating free-form reasoning outputs requires robust answer verification, as model predictions may vary in surface form while being semantically correct. Our validation loop employs a four-stage verification process:

1. **Exact Match.** We first check whether the predicted answer string exactly matches the ground-truth string after normalization (e.g., case-folding and whitespace trimming).

2. **Containment.** If exact match fails, we check whether the normalized gold answer appears as a substring within the model output. This captures predictions where the answer is embedded in additional text.

3. **Token-level F1.** We compute token-level precision, recall, and F1 between the predicted output and the gold answer. Predictions are accepted if the F1 score $\geq 0.90$, ensuring high lexical overlap even under paraphrasing.

4. **Semantic Similarity.** Finally, we compute cosine similarity between SBERT embeddings of the predicted answer and the gold answer. Predictions are marked correct if the similarity score $\geq 0.8$.

A prediction is considered correct if it satisfies *any* of the four criteria. This layered procedure provides robustness to surface-level variation while enforcing semantic fidelity to the ground-truth answer.

For arithmetic datasets, we additionally use the `math_verify` library to parse, simplify, and compare numeric expressions. This ensures that mathematically equivalent forms (e.g., "$\frac{3}{2}$" vs. "1.5") are treated as correct, even if their textual forms differ.

Finally, in our **evaluation experiments** (Section 4), when testing trained models on external benchmarks via the LM Evaluation Harness (Srivastava et al., 2023), we adapt the same four-stage verification method (including thresholds and `math_verify`) to ensure consistency across training validation and benchmark evaluation.

### A.3.5    EVALUATION SETUP

All evaluations are conducted using the LM Evaluation Harness (Srivastava et al., 2023). To ensure consistency with our training validation, we adapt the same four-stage answer verification procedure (Section A.3.4), including thresholds for token-level F1 and semantic similarity, as well as the use of `math_verify` for numeric equivalence checking.

**Multiple-choice tasks.** For benchmarks originally framed as multiple-choice question answering (e.g., StrategyQA, QASC), we convert them into free-form generation tasks. Specifically, we discard option letters and use the text of the correct option as the gold answer. Model outputs are then evaluated against these free-form answers using the verification pipeline.

**Contextual tasks.** For tasks that provide long passages as context (e.g., MuSiQue, OpenBookQA), we extract only the sentences marked as relevant by dataset annotations and provide these as the model's context. This reduces context length while preserving all information necessary to answer the question.

**Metrics.** We report *pass@5* accuracy, where a prediction is considered correct if any of the top-5 generated candidates passes verification. All reported results include the standard error (stderr) across evaluation runs.

### A.4    ADDITIONAL EXPERIMENT RESULTS

Table 7 and table 8 show the accuracies and standard error of the math reasoning and multi-hop reasoning benchmarks.

Table 9 and table 10 show the pass@1 accuracies and standard errors of the math reasoning and multi-hop reasoning benchmarks with model using greedy decoding.

### A.5    REWRITE DEPTH SHIFTS THE DIFFICULTY DISTRIBUTION.

To better understand how rewrite depth affects the structure of the training signal, we analyze the distribution of examples by their minimal number of reasoning steps under different maximum rewrite depths. Table 13 and Figure 5 show that progressively adding deeper rewrites systematically increases the share of low-step (i.e., easier) questions, while reducing the long tail of high-step instances. For example, questions solvable in 0-1 steps rise from 0.1% at depth 0 to 45.1% at depth 4+. Conversely, high-step examples (6+) become increasingly rare as depth increases.

This confirms that rewrite depth does not simply add more training data, but rebalances the effective curriculum: shallow depths preserve a wide difficulty spectrum, while deeper depths concentrate probability mass on easier instances. Combined with Table 4, these findings suggest that moderate rewrite depth is beneficial because it increases exposure to simpler reasoning patterns without collapsing the full difficulty range.

| Methods | GSM8K | AddSub | ASDiv | Multi-Arith | SVAMP |
|---|---|---|---|---|---|
| **OPT-1.3B** | | | | | |
| Base | 3.79±0.53 | 1.83±1.29 | 4.05±0.79 | 2.22±1.10 | 5.69±1.34 |
| CoT KD | 27.75±1.23 | 9.17±2.78 | 20.23±1.62 | 63.33±3.60 | 20.74±2.35 |
| LoT (Ours) | 31.16±1.28 | 33.03±4.53 | 41.75±1.99 | 68.33±3.48 | 38.46±2.82 |
| **OPT-2.7B** | | | | | |
| Base | 3.34±0.49 | 1.83±1.29 | 4.21±0.81 | 3.33±1.34 | 6.69±1.45 |
| CoT KD | 31.01±1.27 | 8.26±2.65 | 26.38±1.77 | 71.11±3.39 | 19.06±2.28 |
| LoT (Ours) | 33.97±1.30 | 40.37±4.72 | 45.95±2.01 | 80.56±2.96 | 44.15±2.88 |
| **Pythia-1.4B** | | | | | |
| Base | 2.96±0.47 | 0.00±0.00 | 4.85±0.87 | 1.67±0.96 | 8.03±1.57 |
| CoT KD | 26.00±1.21 | 3.67±1.81 | 21.36±1.65 | 59.44±3.67 | 19.73±2.31 |
| LoT (Ours) | 28.35±1.24 | 24.77±4.15 | 40.78±1.98 | 63.33±3.60 | 38.46±2.82 |
| **Pythia-2.8B** | | | | | |
| Base | 3.71±0.52 | 1.83±1.29 | 6.63±1.00 | 4.44±1.54 | 9.70±1.71 |
| CoT KD | 33.43±1.30 | 11.93±3.12 | 31.88±1.88 | 74.44±3.26 | 24.08±2.48 |
| LoT (Ours) | 32.98±1.29 | 32.11±4.49 | 46.76±2.01 | 72.78±3.33 | 44.82±2.88 |

Table 7: Pass@5 mean accuracy (%) and standard error on GSM8K test split and four math reasoning benchmarks.

| Methods | EntailmentBank | QASC | OpenBookQA | StrategyQA | MuSiQue |
|---|---|---|---|---|---|
| **OPT-1.3B** | | | | | |
| Base | 22.0±4.16 | 17.2±1.69 | 14.8±1.59 | 32.4±2.10 | 2.20±0.66 |
| CoT KD | 36.0±4.82 | 46.0±2.23 | 47.4±2.24 | 22.6±1.87 | 14.2±1.56 |
| LoT (Ours) | 41.0±4.94 | 52.8±2.23 | 43.6±2.22 | 47.8±2.24 | 39.2±2.19 |
| **OPT-2.7B** | | | | | |
| Base | 21.0±4.09 | 28.8±2.03 | 12.6±1.49 | 33.6±2.11 | 2.20±0.66 |
| CoT KD | 40.0±4.92 | 56.2±2.22 | 46.0±2.23 | 54.0±2.23 | 43.6±2.22 |
| LoT (Ours) | 41.0±4.94 | 60.4±2.19 | 47.8±2.24 | 51.2±2.24 | 24.0±2.12 |
| **Pythia-1.4B** | | | | | |
| Base | 13.0±3.38 | 45.0±2.23 | 34.2±2.12 | 24.4±1.92 | 12.0±1.45 |
| CoT KD | 36.0±4.82 | 55.2±2.23 | 44.6±2.23 | 36.4±2.15 | 42.0±2.21 |
| LoT (Ours) | 39.0±4.90 | 56.6±2.22 | 36.8±2.16 | 53.2±2.23 | 39.2±2.19 |
| **Pythia-2.8B** | | | | | |
| Base | 10.0±3.02 | 39.0±2.18 | 32.4±2.10 | 29.2±2.04 | 18.8±1.75 |
| CoT KD | 32.0±4.69 | 32.2±2.09 | 28.2±2.01 | 55.6±2.22 | 42.2±2.21 |
| LoT (Ours) | 40.0±4.92 | 48.8±2.24 | 37.8±2.17 | 60.0±2.19 | 45.0±2.23 |

Table 8: Pass@5 mean accuracy (%) and standard error on EntailmentBank test split and four multi-hop reasoning benchmarks.

| Methods | GSM8K | AddSub | ASDiv | MultiArith | SVAMP |
|---|---|---|---|---|---|
| **OPT-1.3B** | | | | | |
| Base | 1.14±0.29 | 0.92±0.92 | 0.81±0.36 | 0.56±0.56 | 2.34±0.88 |
| CoT KD | 16.76±1.03 | 6.42±2.36 | 12.62±1.34 | 46.67±3.73 | 10.70±1.79 |
| LoT (Ours) | 17.36±1.04 | 24.77±4.15 | 26.86±1.78 | 47.22±3.73 | 21.40±2.38 |
| **OPT-2.7B** | | | | | |
| Base | 1.14±0.29 | 0.92±0.92 | 1.94±0.56 | 0.56±0.56 | 3.01±0.99 |
| CoT KD | 19.18±1.08 | 4.59±2.01 | 15.70±1.46 | 52.22±3.73 | 9.03±1.66 |
| LoT (Ours) | 21.83±1.14 | 29.36±4.38 | 34.79±1.92 | 53.89±3.73 | 32.78±2.72 |
| **Pythia-1.4B** | | | | | |
| Base | 1.59±0.34 | 0.92±0.92 | 1.29±0.46 | 1.67±0.96 | 1.34±0.67 |
| CoT KD | 13.27±0.93 | 3.67±1.81 | 11.81±1.30 | 37.22±3.61 | 10.03±1.74 |
| LoT (Ours) | 17.06±1.04 | 18.35±3.72 | 28.16±1.81 | 41.11±3.68 | 27.76±2.59 |
| **Pythia-2.8B** | | | | | |
| Base | 1.52±0.34 | 0.00±0.00 | 1.78±0.53 | 1.67±0.96 | 1.67±0.74 |
| CoT KD | 19.79±1.10 | 2.75±1.57 | 19.58±1.60 | 52.22±3.73 | 11.37±1.84 |
| LoT (Ours) | 20.70±1.12 | 23.85±4.10 | 31.55±1.87 | 51.67±3.74 | 30.10±2.66 |

Table 9: Pass@1 mean accuracy (%) and standard error on GSM8K test split and four math reasoning benchmarks using greedy decoding.

| Methods | EntailmentBank | QASC | OpenBookQA | StrategyQA | MuSiQue |
|---|---|---|---|---|---|
| **OPT-1.3B** | | | | | |
| Base | 8.00±2.73 | 1.40±0.53 | 7.60±1.19 | 13.60±1.53 | 0.00±0.00 |
| CoT KD | 16.00±3.68 | 35.00±2.14 | 37.60±2.17 | 9.00±1.28 | 4.20±0.90 |
| LoT (Ours) | 21.00±4.09 | 37.60±2.17 | 32.20±2.09 | 22.00±1.85 | 28.00±2.01 |
| **OPT-2.7B** | | | | | |
| Base | 6.00±2.39 | 13.60±1.53 | 0.80±0.40 | 14.60±1.58 | 0.00±0.00 |
| CoT KD | 22.00±4.16 | 41.00±2.20 | 31.80±2.08 | 19.40±1.77 | 22.40±1.87 |
| LoT (Ours) | 17.00±3.78 | 45.80±2.23 | 34.80±2.13 | 20.60±1.81 | 21.20±1.83 |
| **Pythia-1.4B** | | | | | |
| Base | 4.00±1.97 | 18.60±1.74 | 14.80±1.59 | 5.80±1.05 | 1.60±0.56 |
| CoT KD | 17.00±3.78 | 39.00±2.18 | 31.40±2.08 | 16.20±1.65 | 19.80±1.78 |
| LoT (Ours) | 23.00±4.23 | 37.40±2.17 | 24.00±1.91 | 24.40±1.92 | 24.40±1.92 |
| **Pythia-2.8B** | | | | | |
| Base | 2.00±1.41 | 17.00±1.68 | 11.20±1.41 | 3.40±0.81 | 3.40±0.81 |
| CoT KD | 18.00±3.86 | 19.20±1.76 | 18.60±1.74 | 30.20±2.06 | 15.60±1.62 |
| LoT (Ours) | 28.00±4.51 | 37.80±2.17 | 27.40±2.00 | 28.20±2.01 | 27.80±2.01 |

Table 10: Pass@1 mean accuracy (%) and standard error on EntailmentBank test split and four multi-hop reasoning benchmarks using greedy decoding.

| Depth | GSM8K | AddSub | ASDiv | MultiArith | SVAMP | Average |
|---|---|---|---|---|---|---|
| 0 | 10.46±0.84 | 6.42±2.36 | 9.22±1.16 | 17.78±2.86 | 7.02±1.48 | 10.18 |
| ≤1 | 30.55±1.27 | 26.61±4.25 | 40.61±1.98 | 67.78±3.49 | 27.76±2.59 | 38.66 |
| ≤2 | 28.81±1.25 | 32.11±4.49 | 48.22±2.01 | 71.11±3.39 | 42.81±2.87 | 44.61 |
| ≤3 | 32.07±1.29 | 30.28±4.42 | 47.73±2.01 | 77.22±3.13 | 43.14±2.87 | 46.09 |
| All | 31.16±1.28 | 33.03±4.53 | 41.75±1.99 | 68.33±3.48 | 38.46±2.82 | 42.55 |

Table 11: Math reasoning benchmark performance across reasoning depths. Numbers are pass@5 mean accuracy (%) with standard error.

| Method | GSM8K | AddSub | ASDiv | MultiArith | SVAMP | Average |
|---|---|---|---|---|---|---|
| Random | 29.34±1.25 | 27.52±4.30 | 32.69±1.89 | 66.67±3.52 | 31.77±2.70 | 37.60 |
| Easy→Hard | 28.96±1.25 | 31.19±4.46 | 42.39±1.99 | 61.67±3.63 | 40.47±2.84 | 40.94 |
| Hard→Easy | 21.38±1.13 | 5.50±2.19 | 11.33±1.28 | 56.67±3.70 | 6.69±1.45 | 20.31 |
| Self-evolving | 31.16±1.28 | 33.03±4.53 | 41.75±1.99 | 68.33±3.48 | 38.46±2.82 | 42.55 |

Table 12: Comparison of curriculum strategies on math reasoning benchmarks. Numbers are pass@5 mean accuracy (%) with standard error.

| Reasoning Steps | Depth 0 | Depth 1 | Depth 2 | Depth 3 | Depth 4+ (All) |
|---|---|---|---|---|---|
| 0 | 0 (0.0%) | 2 (0.0%) | 1804 (8.2%) | 4235 (15.3%) | 7320 (22.5%) |
| 1 | 9 (0.1%) | 1880 (12.7%) | 4310 (19.5%) | 6087 (22.0%) | 7352 (22.6%) |
| 2 | 1868 (25.3%) | 4110 (27.9%) | 5734 (26.0%) | 6529 (23.6%) | 6920 (21.2%) |
| 3 | 2124 (28.8%) | 3738 (25.4%) | 4598 (20.8%) | 4940 (17.9%) | 5063 (15.6%) |
| 4 | 1554 (21.1%) | 2443 (16.6%) | 2818 (12.7%) | 2950 (10.7%) | 2989 (9.2%) |
| 5 | 951 (12.9%) | 1371 (9.3%) | 1542 (7.0%) | 1592 (5.8%) | 1601 (4.9%) |
| 6 | 451 (6.1%) | 654 (4.4%) | 729 (3.3%) | 745 (2.7%) | 745 (2.3%) |
| 7 | 241 (3.3%) | 333 (2.3%) | 354 (1.6%) | 355 (1.3%) | 355 (1.1%) |
| 8 | 105 (1.4%) | 135 (0.9%) | 137 (0.6%) | 137 (0.5%) | 137 (0.4%) |
| 9 | 45 (0.6%) | 50 (0.3%) | 52 (0.2%) | 52 (0.2%) | 52 (0.2%) |
| 10 | 16 (0.2%) | 17 (0.1%) | 19 (0.1%) | 19 (0.1%) | 19 (0.1%) |
| 11 | 3 (0.0%) | 6 (0.0%) | 6 (0.0%) | 6 (0.0%) | 6 (0.0%) |
| 12 | 3 (0.0%) | 3 (0.0%) | 3 (0.0%) | 3 (0.0%) | 3 (0.0%) |
| 13 | 1 (0.0%) | 1 (0.0%) | 1 (0.0%) | 1 (0.0%) | 1 (0.0%) |
| 14 | 1 (0.0%) | 2 (0.0%) | 2 (0.0%) | 2 (0.0%) | 2 (0.0%) |
| 15 | 1 (0.0%) | 1 (0.0%) | 1 (0.0%) | 1 (0.0%) | 1 (0.0%) |
| **Total** | 7373 (100%) | 14746 (100%) | 22110 (100%) | 27654 (100%) | 32566 (100%) |

Table 13: Distribution of minimal reasoning steps by maximum rewrite depth. Percentages are computed column-wise relative to each depth total. Deeper rewrite depths increasingly skew the distribution toward low-step (easier) questions.

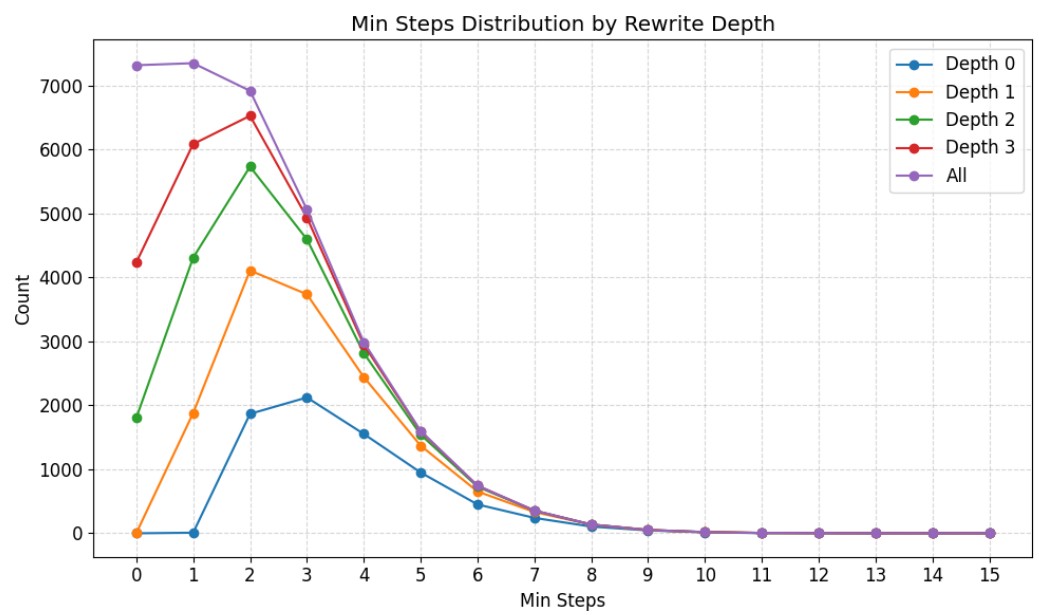

Figure 5: Distribution of minimal reasoning steps under different rewrite depths.

A.6 ABLATION: EMPIRICAL DIFFICULTY VS. STEP-BASED DIFFICULTY.

We first examine whether the step-based difficulty measure aligns with empirical difficulty. Figure 6 plots the average model success rate against minimal step count for three models of different sizes. We observe a clear negative correlation between the required number of reasoning steps and empirical success rate: lower-step problems are consistently easier across models, while higher-step problems are more frequently failed. Motivated by this correlation, we construct an alternative curriculum that replaces step-based difficulty buckets with empirical difficulty buckets derived directly from the Qwen2.5-7B model.

Each training example is assigned to one of five buckets based on its observed success rate: from bucket 0 (success $\geq 0.8$) to bucket 4 (success $< 0.2$). The resulting data distribution is shown in Table 14. We then train a curriculum model using the same training setup as LoT, but sampling examples from empirical buckets instead. Final evaluation across in-domain and OOD benchmarks is presented in Table 15.

| Bucket | Success Range | Count (%) |
|--------|---------------|-----------|
| 0 | $\geq 0.8$ | 13,558 (40.83%) |
| 1 | $[0.6, 0.8)$ | 6,668 (20.08%) |
| 2 | $[0.4, 0.6)$ | 5,213 (15.70%) |
| 3 | $[0.2, 0.4)$ | 4,179 (12.59%) |
| 4 | $< 0.2$ | 3,587 (10.80%) |
| **Total** | — | 33,205 (100%) |

Table 14: Data distribution under empirical difficulty bucketing based on Qwen2.5–7B success rate estimates.

| Methods | GSM8K | ASDiv | AddSub | MultiArith | SVAMP |
|---------|-------|-------|--------|------------|-------|
| Base | 21.61±1.13 | 11.17±1.27 | 5.50±2.19 | 15.00±2.67 | 10.70±1.79 |
| CoT KD | 73.84±1.21 | 77.67±1.68 | 50.46±4.81 | 98.33±0.96 | 60.54±2.83 |
| LoT | **74.60±1.20** | **74.92±1.75** | **70.64±4.38** | **98.89±0.78** | **71.57±2.61** |
| Success-rate LoT | 63.68±1.32 | 57.77±1.99 | 64.22±4.61 | 59.44±3.67 | 46.49±2.89 |

Table 15: Pass@5 accuracy (%) for curricula constructed using empirical difficulty bucketing.

While empirical difficulty correlates with minimal step count, the success-rate curriculum performs markedly worse than LoT. One explanation is that empirical difficulty mixes structural difficulty with surface-level distribution artifacts, leading to uneven exposure across reasoning patterns. In contrast, step-based difficulty explicitly targets inferential depth, producing a more balanced ladder of abstractions. These results suggest that while empirical hardness is partially aligned with step complexity, it is a weaker organizing principle for reasoning curricula than step-based progressive rewrites.

A.7 ABLATION: EFFECT OF REWRITER MODEL SCALE

In this section we examine whether the scale of the rewrite model matters. We generate LoT rewrites using three Qwen2.5 Instruct models of increasing size (7B, 14B, 72B) and vary whether the rewriter is also used as the teacher model or whether the teacher is held fixed. In the *rewriter-as-teacher* setting, the same model provides both the rewritten training instances and the reference reasoning traces. In the *fixed-teacher* setting, each rewriter only produces rewritten questions, while a single teacher model (GPT-5-mini) provides the answers and reasoning traces for all variants. Table 16 summarizes the number of rewrites produced under each configuration.

We then train Qwen2.5-7B using these rewritten datasets and evaluate across in-distribution and OOD benchmarks. Results are shown in Table 17.

Our ablation results indicate that LoT is sensitive to the capability of the rewriting model. Stronger and more instruction-following rewriters (e.g., GPT-5-mini) generate more accurate and semantically consistent progressive rewrites, which leads to better downstream performance. We

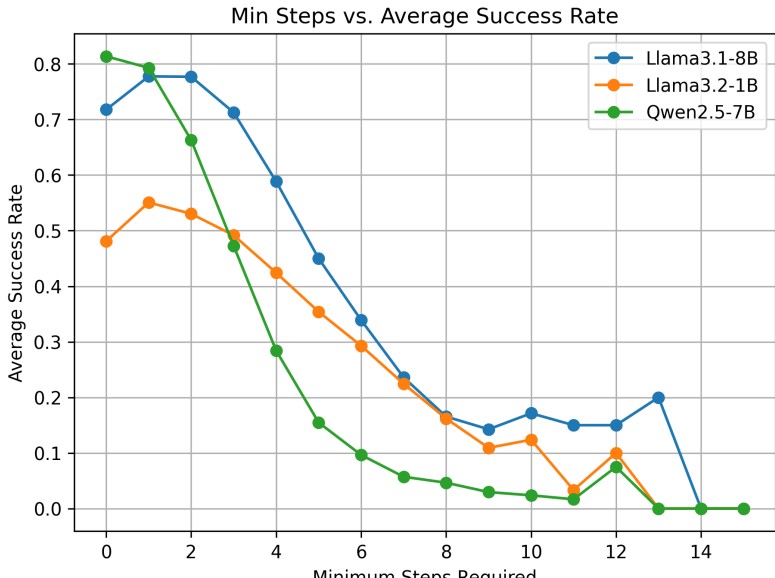

Figure 6: Average model success rate as a function of minimal reasoning steps. Lower-step examples are generally easier across three model families.

| Rewriter Model | Original Questions | Rewrite Questions | Train Size | Val Size |
|---|---|---|---|---|
| Qwen2.5-7B-Instruct | 7373 | 17611 | 24984 | 635 |
| Qwen2.5-14B-Instruct | 7373 | 23028 | 30401 | 623 |
| Qwen2.5-72B-Instruct | 7372 | 22335 | 29707 | 616 |
| GPT-5-mini | 7373 | 25193 | 32566 | 639 |

Table 16: Dataset statistics when using Qwen2.5 models of different sizes to generate LoT rewrites.

| Rewriter | Teacher | GSM8K | AddSub | ASDiv | MultiArith | SVAMP |
|---|---|---|---|---|---|---|
| Qwen2.5-7B-Instr. | Qwen2.5-7B-Instr. | 17.66±1.05 | 70.64±4.38 | 69.74±1.85 | 30.56±3.44 | 58.86±2.85 |
| Qwen2.5-14B-Instr. | Qwen2.5-14B-Instr. | **22.06**±1.14 | **84.40**±3.49 | **73.95**±1.77 | **37.78**±3.62 | **63.88**±2.78 |
| Qwen2.5-72B-Instr. | Qwen2.5-72B-Instr. | 16.68±1.03 | 65.14±4.59 | 67.80±1.88 | 28.33±3.37 | 54.18±2.89 |
| Qwen2.5-7B-Instr. | GPT-5-mini | **25.70**±1.20 | **78.90**±3.93 | 71.04±1.83 | **30.56**±3.44 | 54.85±2.88 |
| Qwen2.5-14B-Instr. | GPT-5-mini | 19.94±1.10 | 77.06±4.05 | 72.82±1.79 | 30.00±3.43 | **62.88**±2.80 |
| Qwen2.5-72B-Instr. | GPT-5-mini | 11.75±0.89 | 31.19±4.46 | 34.95±1.92 | 16.67±2.79 | 23.08±2.44 |
| GPT-5-mini | GPT-5-mini | 74.60±1.20 | 70.64±4.38 | 74.92±1.75 | 98.89±0.78 | 71.57±2.61 |

Table 17: Performance of Qwen2.5-7B trained using rewrites generated by Qwen2.5 models of different sizes. A fixed-teacher setting (GPT-5-mini) and a rewriter-as-teacher setting are both shown.

observe that when Qwen models are used as both rewriter and teacher, performance generally lags behind GPT-5-mini, reflecting differences in rewrite quality.

When keeping GPT-5-mini as the teacher but using Qwen models as rewriters, only the Qwen-7B rewriter shows improvement over its rewriter-as-teacher configuration. Qwen-14B remains comparable, while Qwen-72B degrades significantly. This suggests that the rewriter has a stronger influence than the teacher on downstream performance: if the rewrites introduce semantic inconsistencies, the curriculum becomes less helpful regardless of teacher strength.

Overall, these results show that LoT benefits from higher-quality rewriting models, but continues to outperform standard CoT distillation even when the rewriter is significantly weaker. This

dependency is consistent with broader observations in model-generated data pipelines, where data quality strongly conditions downstream performance.

## A.8 Progressive Rewrite Prompts

To construct progressive difficulty ladders, we prompted a rewrite model with carefully designed instructions. Below we include the exact prompts used for each dataset to ensure reproducibility.

### A.8.1 EntailmentBank Prompt

```
You are an expert at reasoning question simplification.
I will provide you with a reasoning problem in JSON format that
contains:

- "instruction": the solving instruction
- "input": the context and question
- "output": the reasoning chain and final answer

Your task is to automatically generate a progressive difficulty ladder
of simplified versions of this problem.
Each new version should make the reasoning easier by moving more
intermediate conclusions (from the reasoning steps in the output)
directly into the input context.
Stop when the problem has become trivial (e.g., the final hypothesis
is already in the input).
For each version, also output the minimum number of reasoning steps
required to reach the final answer from that version's input.
Treat a reasoning step as a necessary inferential move that derives a
new statement from previous facts/conclusions (e.g., one arithmetic
operation, one logical implication, one factual lookup from the
provided context).
Count merged paraphrases/restatements as 0 additional steps; do not
double-count trivially equivalent rewrites.
When multiple independent sub-derivations are needed before a final
combination, count each indispensable sub-derivation as one step.
The count must be a non-negative integer; use 0 for a trivial version
where the answer is directly stated in the input.
Ensure monotonic non-increase across versions (later versions should
never require more steps than earlier ones).

Guidelines:

1. Identify all intermediate conclusions (int1, int2, ...) in the
original reasoning chain.
2. Create Version 1 as the original (no added intermediates).
3. Then generate subsequent versions, each time inserting one or more
intermediates into the input.
4. You may decide the number of versions automatically | fewer if the
chain is short, more if it is long.
5. For each version, output in a fenced JSON code block with the
following keys:
   - "instruction"
   - "input"
   - "answer" (string, the final answer to the problem)
   - "reasoning" (string, the reasoning chain leading to the answer)
   - "min_steps" (integer, the minimum number of steps to reach the
   answer)
   - "min_steps_note" (a short explanation explaining the count)
6. Precede each block with a Markdown label like:
   ## Version N | [difficulty descriptor]
   Then immediately follow with:
   ```json
   { ... }
   ```
```

Goal: produce a set of progressively easier problems, where the solver
needs fewer reasoning steps at each level, and report the minimum
required steps for each version.

### A.8.2 GSM8K PROMPT

You are an expert at math word problem simplification.
I will provide you with a math problem in JSON format that contains:

– "question": the text of the problem
– "answer": the worked-out reasoning and final numeric answer

Your task is to automatically generate a *progressive difficulty ladder*
of simplified versions of this problem.
Each new version should make the reasoning easier by moving more
intermediate results (from the solution steps in the answer) directly
into the problem statement.
Stop when the problem has become trivial (e.g., the final numeric answer
is already stated in the problem).

For each version, also output the **minimum number of reasoning steps**
required to reach the final answer from that version's problem statement.
– Treat a *reasoning step* as a necessary mathematical operation or
logical inference (e.g., one arithmetic operation, one fraction
simplification, one comparison).
– Do not double-count trivial rewrites or restatements.
– When multiple sub-calculations are required before combining, count
each indispensable sub-calculation as one step.
– The count must be a non-negative integer; use **0** when the answer
is already stated in the problem.
– Ensure the counts are **monotonic non-increasing** across versions
(later versions should never require more steps than earlier ones).

Guidelines:

1. Identify all intermediate results (e.g., partial sums,
multiplications, divisions) in the original worked-out solution.
2. Create **Version 1** as the original (no added intermediates).
3. Then generate subsequent versions, each time inserting one or more
intermediate results directly into the problem statement.
4. You may decide the number of versions automatically | fewer if the
chain is short, more if it is long.
5. For each version, output in a fenced JSON code block with the
following keys:
   – "question" (string, the modified problem statement)
   – "answer" (string, the final numeric answer only)
   – "reasoning" (string, the reasoning steps leading to the answer)
   – "min_steps" (integer, the minimum number of steps required)
   – "min_steps_note" (short explanation for the count)
6. Precede each block with a Markdown label like:
   ## Version N | [difficulty descriptor]
   Then immediately follow with:
   ```json
   { ... }
   ```

Goal: produce a set of progressively easier GSM8K problems, where the
solver needs fewer reasoning steps at each level, and report the minimum
required steps for each version.