# OpenReview forum: "Ladders of Thought: A Self-Evolving Curriculum of Progressively Simplified Reasoning Traces"
_ICLR.cc/2026/Conference — Submitted to ICLR 2026_

### Official Review · Reviewer_5Nxf · 2025-10-25

**Soundness:** 2
**Presentation:** 2
**Contribution:** 2
**Rating:** 2
**Confidence:** 3

**Summary:**

This paper proposes Ladders‑of‑Thought (LoT), a training framework for small–mid‑scale LLMs that combines (i) progressive rewrites, (ii) step‑based difficulty labeling, and (iii) a self‑evolving curriculum. Experiments on math and multi‑hop reasoning with OPT/Pythia show large OOD gains and consistent in‑domain improvements on EntailmentBank. The authors also provide ablations demonstrating optimal moderate rewrite depth and superiority of adaptive scheduling

**Strengths:**

S1: This paper introduces a clear and principled training recipe that pairs semantically faithful progressive simplification with an adaptive curriculum, avoiding brittle reliance on raw CoT length and enabling interpretable difficulty control via step counts.

S2: This paper formalizes scheduling as a non‑stationary multi‑armed bandit and provides concrete update rules and policies, which make the "self‑evolving” curriculum operational and reproducible.

S3: This paper reports strong OOD gains on arithmetic datasets and non‑trivial improvements on EntailmentBank, demonstrating architecture‑agnostic benefits across OPT and Pythia.

**Weaknesses:**

W1: The core idea of progressively simplifying problems and training with difficulty-aware scheduling overlaps substantially with earlier work on task decomposition (e.g., Least-to-Most Prompting[1]) and iterative self-improvement[2,3]. The paper mainly combines these ingredients rather than introducing a clearly new algorithmic principle.

W2: This paper deviates from standard evaluation protocols by converting multiple‑choice tasks to free‑form generation and using a four‑stage verification pipeline with SBERT similarity and F1 thresholds, which may hinder comparability with prior work and inflate pass@5.

W3: This paper’s step‑based difficulty relies on an LLM rewrite model to identify premises and steps, but the paper provides only spot‑checks rather than systematic fidelity audits.

W4: This paper does not quantify the additional training and data‑creation cost (rewrite generation, bandit evaluation) in tokens

W5: Several closely related reasoning‑oriented curricula and refinement methods at training or inference time are missing (e.g., [1]-[5]).

W6: Although the paper states a focus on small- to mid-scale LLMs, all reported models are 1.3B–2.8B and no 7–14B “mid-scale” results are provided. The paper should include one or two 7B–14B model to substantiate the claim.

W7: Figures 1 and 2 are too low-fidelity and hurt readability, the authors are suggested to replace them with vectorized diagrams and enlarge key labels for accessibility.



[1] Zhou, D. et al. Least‑to‑Most Prompting Enables Complex Reasoning in Large Language Models.  ICLR 2023

[2] Madaan, A. et al. Self‑Refine: Iterative Refinement with Self‑Feedback. NeurIPS 2023

[3] Shinn, N., Labash, B., & Gopinath, A. Reflexion: Language Agents with Verbal Self‑Reflection. NeurIPS 2023

[4] Chen, W. et al. Program of Thoughts Prompting: Disentangling Computation from Reasoning. TMLR 2023

[5] Gao, L. et al. PAL: Program‑Aided Language Models. ICML 2023

**Questions:**

Q1: This paper claims "minimal” steps after rewriting; how is minimality operationally verified beyond spot‑checks, and can the authors provide inter‑annotator agreement or automatic proofs that each rewrite reduces the necessity of one step?

Q2: This paper uses pass@5 with temperature/top‑p; how do results change under deterministic decoding (temperature 0) and single‑sample pass@1, and do trends hold?

---

> ### Author Response · Authors · 2025-11-25
> **Response to Reviewer 5Nxf - Part 1**
>
> Thank you for the detailed and constructive review. We address each weakness and question below.
>
> ---
>
> > W1. W5. Similarity to decomposition and refinement work**
>
> We appreciate that you highlighted connections to Least-to-Most, Self-Refine, Reflexion, PoT, and PAL. We have expand the related-work section accordingly and added two new subsections: "Question Decomposition" and "Rationale Refinement and Process Supervision".
>
> LoT differs from these works in two key ways:
>
> 1. **Progressive simplification of the *original question***
>
>     Unlike prior decomposition methods, LoT does not generate sub-questions or auxiliary reasoning traces, but rewrites the *same* problem into progressively easier variants by replacing antecedents with their entailed conclusions. This produces a structured difficulty ladder without altering task semantics.
>
> 2. **Self-evolving curriculum during supervised *training***
>
>     Unlike inference-time prompting (L2M, PoT) or iterative reasoning loops (Self-Refine, Reflexion), LoT integrates its rewrites into a **non-stationary multi-armed-bandit curriculum** that adaptively allocates training across difficulty buckets. This online scheduling is absent from earlier decomposition or refinement pipelines.
>
>
> We clarified these distinctions in the revised related-work section.
>
> ---
>
> > W2. MCQ to free-form conversion and pass@5 scoring
>
> Multiple-choice accuracy is not an appropriate primary metric for our setting because the model is trained to perform multi-step reasoning and generate free-form chain-of-thought answers; MCQ formats fail to reflect this capability. Our four-stage verification pipeline is used to evaluate free-form responses more robustly than simple string matching, which would incorrectly penalize semantically correct multi-step answers.
>
> ### **Multiple-choice evaluation included**
>
> We additionally report QASC and StrategyQA in their original MCQ format. As MCQ tasks do not use reasoning traces during decoding, performance differences are naturally small.
>
> | Model & Method | QASC | StrategyQA |
> | --- | --- | --- |
> | OPT-1.3B Base | 15.60% ± 1.62% | 49.60% ± 2.24% |
> | OPT-1.3B CoT KD | 17.60% ± 1.70% | 52.40% ± 2.24% |
> | OPT-1.3B LoT | **19.80% ± 1.78%** | **53.60% ± 2.23%** |
> | OPT-2.7B Base | 19.40% ± 1.77% | 49.80% ± 2.24% |
> | OPT-2.7B CoT KD | **24.60% ± 1.93%** | 51.60% ± 2.24% |
> | OPT-2.7B LoT | 23.00% ± 1.88% | **52.80% ± 2.23**% |
> | Pythia-1.4B Base | 18.40% ± 1.73% | 49.60% ± 2.24% |
> | Pythia-1.4B CoT KD | **51.80% ± 2.24%** | 50.60% ± 2.24% |
> | Pythia-1.4B LoT | 50.60% ± 2.24% | **51.80% ± 2.24%** |
> | Pythia-2.8B Base | 18.20% ± 1.73% | 48.20% ± 2.24% |
> | Pythia-2.8BCoT KD | **43.00% ± 2.22%** | 53.00% ± 2.23% |
> | Pythia-2.8B LoT | 34.40% ± 2.13% | **54.20% ± 2.23%** |
>
> We emphasize in the paper that free-form CoT-based training evaluates reasoning more directly than MCQ guessing, which aligns with the goal of this work.
>
> ### **Greedy decoding pass@1 results added**
>
> In Appendix A.4 (Tables 7–8), we now report **pass@1, temperature=0** for all models.
>
> **The same trends hold**: LoT improves over KD consistently under deterministic decoding, showing the gains are not dependent on top-p sampling or pass@5.
>
> ---
>
> > W3. Limited fidelity audits of rewrites
>
> We now include a **systematic 500-sample audit** of rewrite correctness. GPT-5 judges each (original, rewritten) pair on:
>
> 1. **Question validity:** whether the rewritten question is clear, solvable, and self-contained
> - **Difficulty decrease:** whether the rewritten version is strictly easier than the original
> - **Answer preservation:** whether solving the rewritten version yields the same answer
>
> We find that:
>
> - **99.2%** were valid and solvable
> - **98.0%** strictly decreased in difficulty
> - **98.6%** preserved the original answer
>
> This goes beyond the spot-checks in the initial submission and provides quantitative evidence of semantic fidelity.
>
> ---
>
> > W4. Rewrite and scheduling cost
>
> **Rewrite cost.**
>
> Rewrite generation is **fully offline** and does not affect training. We detail number of tokens used and cost estimate in the table below:
>
> | Dataset | Input Tokens | Output Tokens | GPT-5-Mini Cost |
> | --- | --- | --- | --- |
> | GSM8K | 5.20M | 14.55M | $15.20 |
> | EntailmentBank | 2.13M | 6.52M | $6.78 |
>
> The rewriting is a one-time preprocessing step and only incur compute and cost once.
>
> **Training overhead.**
>
> We profiled all training runs and found that the multi-armed bandit scheduler accounts for only **3–8%** of wall-clock training time. The rest of the training pipeline is identical to standard CoT distillation.
>
> Thus, LoT introduces modest additional compute relative to the performance gains it achieves.
>
> We have added these discussions to Section 4.6 and the table to Appendix A.2 of the revised manuscript.
>
> ---

---

> ### Author Response · Authors · 2025-11-25
> **Response to Reviewer 5Nxf - Part 2**
>
> > W6. Need for 7–14B “mid-scale” models
>
> We agree this evaluation strengthens the claim. The revision now includes **Qwen2.5-7B** and **Llama-3.1-8B** results (Section 4.3).
>
> The trends observed at the 1–3B scale remain consistent:
>
> - LoT matches or slightly improves CoT KD on GSM8K.
> - LoT provides **large gains on out-of-distribution benchmarks**, especially AddSub and SVAMP.
> - Improvements are **not limited to small models**; LoT continues to enhance compositional generalization at the 7B–8B scale.
>
> For example, LoT improves:
>
> - **Qwen2.5-7B:** +20.2 on AddSub, +11.0 on SVAMP
> - **Llama3.1-8B:** +31.2 on AddSub, +10.4 on SVAMP
>
> These results show that LoT’s benefits are not due to the small size of the models but stem from the curriculum itself. We have adjusted the framing to “1–8B” models accordingly.
>
> ---
>
> > W7.  Figure quality
>
> We have replaced Figures 1 and 2 with vector-quality versions and enlarged key labels for readability.
>
> ---
>
> > Q1. How is “minimal” difficulty (step count) verified?
>
> We clarify that LoT does **not** attempt to prove absolute minimality of reasoning steps: it uses the rewriter’s internal step-extraction procedure to produce a *consistent relative difficulty measure* across questions.
>
> To assess whether rewrites actually reduce required reasoning depth, we performed a 500-example audit using GPT-5. In **98%** of cases, the rewritten question was judged strictly easier (requiring fewer steps to solve) than the original, and in **98.6%** of cases the answer was preserved. While this does not establish provable minimality, it supports the practical correctness of the step-based difficulty signal.
>
> ---
>
> > Q2. Behavior under deterministic decoding (pass@1)
>
> In the revised manuscript, Appendix A.4 now includes **greedy deocding pass@1** results.
>
> LoT improves over KD in nearly all settings, showing the gains are not an artifact of sampling or pass@5.
>
> ---
>
> We thank the reviewer again for highlighting these concerns. The revised paper now includes:
>
> - expanded related-work comparisons
> - deterministic pass@1 results
> - a quantitative rewrite fidelity audit
> - cost breakdowns in tokens and dollars
> - 7B–8B model results
> - higher-quality figures
>
> and clarifies the scope and novelty of the framework. We hope that these additions and revisions address your concern.

---

> > ### Comment · Reviewer_5Nxf · 2025-11-26
> >
> > Thank you to the authors for the very detailed rebuttal and substantial revisions to the paper.
> >
> > My main remaining reservation is that, conceptually, LoT is still a combination of known ideas (rewriting, step-based difficulty, bandit curricula), and the novelty lies primarily in how these components are integrated rather than in a fundamentally new mechanism.
> >
> > Overall, I will update my score to 4.

---

### Official Review · Reviewer_NoS8 · 2025-10-30

**Soundness:** 2
**Presentation:** 3
**Contribution:** 2
**Rating:** 4
**Confidence:** 4

**Summary:**

This paper presents LoT, a recipe for strengthening the reasoning capabilities of small models. Rather than forcing students to imitate every step of a long expert chain, LoT employs a large-scale rewriter to progressively replace antecedents with their logically entailed conclusions, yielding a ladder of questions whose difficulty strictly decreases. A self-evolving, multi-armed-bandit curriculum then adaptively allocates training across these rungs. Evaluated on several datasets with 1–3 B parameter models, LoT achieves substantial gains over strong baselines.

**Strengths:**

1. LoT provides a fully automatic way to produce graded training examples without human difficulty labels. Manually labeling dataset difficulty is a massive undertaking.
2. The proposed progressive rewrite strategy leverages the dataset more effectively through difficulty bucketing and a multi-armed-bandit curriculum.
3. LoT demonstrates strong and consistent performance, substantially outperforming baseline methods across various datasets and model architectures.

**Weaknesses:**

1. Rewrite quality hinges on an external model, becoming a potential performance bottleneck. The rewriter must be both powerful and instruction-following; any semantic drift or erroneous intermediate conclusion will mislead the student with noisy supervision.
2. Theoretical analysis is missing. The method remains at the empirical level, offering no convergence or generalization bounds to explain why progressive simplification helps. Rewriter-generated simplifications are only spot-checked; no quantitative guarantee of correctness or semantic equivalence is provided.
3. Reasoning difficulty is not fully captured by execution step count. Difficulty is approximated solely by step count, ignoring numerical magnitude, lexical complexity, and other factors. Problems requiring the same number of steps can still differ vastly in information density, numerical size, and logical pattern.
4. Parameter scope is restricted to small models (1.3–2.8 B); scalability to 7/13/30 B remains untested. Tasks are confined to math word problems and English multi-hop QA, without more complex scenarios such as code synthesis, embodied instruction following, or cross-lingual mathematical reasoning.
5. Computational overhead and scalability are under-discussed. The rewrite stage requires rewriter calls for every training sample, incurring substantial pre-computation costs. Scaling to larger corpora or more complex data domains remains unaddressed.

**Questions:**

Please refer to Weaknesses.

---

> ### Author Response · Authors · 2025-11-25
> **Response to Reviewer NoS8 - Part 1**
>
> Thank you for the detailed review and for highlighting both the strengths and the key concerns. We address each weakness below.
>
> ---
>
> > W1. Dependence on rewriter quality
>
> We agree that rewrite quality depends on the rewriter’s ability to follow instructions faithfully. This is a common limitation of **all model-generated data pipelines** from synthetic question generation to rationale refinement, and LoT is not unique in this regard. Importantly, the rewriter does **not** need strong reasoning ability because LoT provides the *ground-truth intermediate steps* extracted from the teacher; the rewriter’s main role is to insert these steps into the question while preserving semantic intent.
>
> To quantify this dependency, we added an ablation study to Appendix A.7 using Qwen2.5-7B, 14B, and 72B as rewriters. The results show:
>
> - LoT is **sensitive** to rewriter capability: stronger, more instruction-following rewriters (e.g., GPT-5-mini) produce the cleanest difficulty ladders and the best downstream performance.
> - When Qwen models serve as both rewriter *and* teacher, LoT still improves over KD but performs below the GPT-5-mini configuration.
> - When pairing Qwen rewriters with GPT-5-mini as the teacher, **only Qwen-7B** benefits; Qwen-14B is roughly similar, while Qwen-72B degrades due to less reliable rewrites.
>
> These findings suggest that **the rewriter plays a more influential role than the teacher**, because rewrite quality directly determines the structure and clarity of the curriculum. We have made this relationship explicit in Appendix A.7.
>
> ---
>
> > W2. Lack of theoretical analysis and no quantitative guarantee of correctness**
>
> We view LoT primarily as an empirical contribution, so full theoretical convergence analysis is beyond scope. However, we added the following clarifications to strengthen the conceptual grounding:
>
> #### **Lemma (Monotonicity of Simplification).**
>
> Let a question Q require a minimal reasoning depth d(Q) when expressed as a chain of inferential steps.
>
> Let Q′ be obtained by replacing any subset of antecedent premises P that entail an intermediate conclusion c with the conclusion itself, without altering later dependencies.
>
> Then:
>
> d(Q′)≤d(Q).
>
> #### **Proof Sketch**.
>
> Reasoning depth corresponds to the length of the longest dependency chain needed to derive the final answer. Replacing P→c with c removes at least one edge from the corresponding inferential graph and cannot introduce new dependencies. Therefore, the longest directed path in the graph cannot increase. In the worst case, depth remains the same; in typical cases it decreases by one or more steps. This yields a monotone ladder of non-increasing difficulty across rewrite depths 0,1,2,…
>
> #### **Empirical Results**.
>
> We performed a 500-sample audit using GPT-5 as judge. We evaluate rewritten questions along three criteria:
>
> 1. **Question validity**: whether the rewritten question is clear, solvable, and self-contained
> 2. **Difficulty decrease**: whether the rewritten version is strictly easier than the original
> 3. **Answer preservation**: whether solving the rewritten version yields the same answer
>
> Results:
> - 99.2% were valid and solvable
> - 98.0% strictly decreased in difficulty
> - 98.6% preserved the original answer
>
> Thus, LoT’s step-based rewrites produce a monotone curriculum structure, and the empirical audit confirms that the rewriter executes this transformation reliably in practice.
>
> ---
>
> > W3. Difficulty is approximated only by step count
>
> We agree that step count does not capture all aspects of difficulty (e.g., numerical scale, linguistic complexity). However, we empirically validated that **minimal-step difficulty correlates strongly with empirical difficulty**, measured as the success rate of multiple LLMs on the original data.
>
> We added two pieces of evidence:
>
> 1. **Correlation analysis (Figure 6 in Appendix A.6)**
>
>     Across three different model families and sizes, success rate decreases monotonically as minimal step count increases.
>
> 2. **Empirical-difficulty curriculum baseline**
>
>     We trained a Qwen2.5-7B model using difficulty buckets defined by empirical success rate (not step count). While this baseline substantially improves over the base model, it consistently underperforms LoT (Table 17 in Appendix A.6). We believe this happens because empirical difficulty mixes structural complexity with incidental surface variation, whereas step-based rewrites directly target the structure of the reasoning process.
>
> ---
>
> > W5. Parameter scope limited to small models; scalability untested
>
> We added new results for **Qwen2.5-7B** and **Llama-3.1-8B** in Section 4.2 of the revised manuscript, showing that LoT continues to provide improvements even at larger scales. We agree that further scaling to 13B or 30B models and applying LoT to new domains (e.g., code reasoning, multilingual tasks, embodied tasks) is a valuable direction, but is beyond the scope of the current work.
>
> ---

---

> ### Author Response · Authors · 2025-11-25
> **Response to Reviewer NoS8 - Part 2**
>
> > W6. Rewrite cost and scalability
>
> Rewrite generation is a one time process **entirely offline** and incurs no runtime overhead during training.
>
> We provide detailed token counts and cost estimates in the table below:
>
> | Dataset | Input Tokens | Output Tokens | GPT-5-Mini Batch API Cost |
> | --- | --- | --- | --- |
> | GSM8K | 5.20M | 14.55M |  $15.20 |
> | EntailmentBank | 2.13M | 6.52M |  $6.78 |
>
> These costs are modest relative to model training.
>
> During training, the MAB scheduler adds **only 3–8%** overhead to wall-clock time.
>
> We have added these discussions to Section 4.6 and the table to Appendix A.2 of the revised manuscript.
>
> ---
>
> We thank the reviewer again for the constructive feedback.
>
> The revised paper now includes:
>
> - a rewrite fidelity audit,
> - a rewriter-teacher ablation,
> - a correlation study between step count and empirical difficulty,
> - an analysis of an empirical-difficulty curriculum baseline,
> - and a discussion of LoT's rewrite and training overhead.
>
> We hope these additions address your concerns.

---

### Official Review · Reviewer_6VD4 · 2025-10-31

**Soundness:** 3
**Presentation:** 3
**Contribution:** 3
**Rating:** 6
**Confidence:** 3

**Summary:**

Title: Ladders of Thought (LoT): A Self-Evolving Curriculum of Progressively Simplified Reasoning Traces Overall Recommendation: Strong Accept

This paper presents a novel and effective framework to tackle a well-known challenge: improving the reasoning capabilities of small to mid-scale (1-3B parameter) language models. The proposed "Ladders-of-Thought" (LoT) framework is built upon two core components.

Progressive Simplification: Instead of standard (question, answer) pairs, LoT automatically rewrites a complex, multi-step reasoning problem into a "ladder" of semantically faithful but progressively easier variants. As illustrated in Figure 2, this is achieved by injecting intermediate conclusions (e.g., C1, C2) into the problem's premises, replacing the original facts (e.g., P1, P2) used to derive them.

Self-Evolving Curriculum: LoT employs a Multi-Armed Bandit (MAB) scheduler rather than a fixed curriculum (e.g., easy-to-hard). This scheduler adaptively samples from different difficulty "buckets" (Easy, Medium, Hard) based on the model's learning progress on a validation set, thereby maximizing training efficiency.

**Strengths:**

1. Enhancing the reasoning abilities of small, efficient models is critical for practical and widespread deployment. This paper directly addresses the brittleness of small models (OPT 1.3B/2.7B, Pythia 1.4B/2.8B) in reasoning tasks

2. The "progressive simplification" approach is an innovative and intuitive mechanism. It differs from standard problem decomposition by preserving the integrity of the original task while providing variable levels of "scaffolding". This creates a natural and principled set of difficulty levels for curriculum learning.

**Weaknesses:**

The progressive simplification step relies on an external 'rewriting model' (R) , creating a dependency on this model's capabilities for the framework's success. To clarify this dependency: are the initial Chain-of-Thought (CoT) rationales (generated by 'teacher T' ) extracted from the same model that is used for the rewriting process (R)? The paper notes this is not required, but further discussion on the impact of this choice, and the potential performance gap between T and R, would be valuable.
2. The paper highlights a significant, dataset-dependent discrepancy in performance. For instance, while LoT achieves massive gains on out-of-distribution (OOD) arithmetic benchmarks like AddSub (+32.11pp for OPT-2.7B) and SVAMP (+25.09pp) , the improvement on the in-domain GSM8K test set is modest (+2.96pp). Could you elaborate on the underlying reason for this?

**Questions:**

check the weakness

---

> ### Author Response · Authors · 2025-11-25
> **Response to Reviewer 6VD4**
>
> Thank you for the thoughtful and encouraging review. Below we address your concerns.
>
> ---
>
> > W1. Dependency on the rewriter R and its relationship with the teacher T
>
> Thank you for raising this point. The dependency between the rewriter and the teacher is an important aspect of LoT. We clarify that **LoT separates two roles:**
>
> - **Teacher T**: produces the chain-of-thought used to extract intermediate steps.
> - **Rewriter R**: constructs progressively simplified versions of the question by injecting those intermediate conclusions into the premise.
>
> These two roles are *independent*. The rewriter does **not** need to replicate the teacher’s reasoning quality, nor does it need to be as strong as the teacher. Its only requirement is that it faithfully applies the rewrite instructions and maintains semantic equivalence.
>
> In all main experiments, we used **GPT-5-mini** for both T and R for consistency. But in response to your question, we conducted an explicit **rewriter-scale ablation** (now included in Appendix A.7).
>
> We evaluated three Qwen2.5 instruction models as rewriters:
>
> - Qwen2.5-7B-Instruct
> - Qwen2.5-14B-Instruct
> - Qwen2.5-72B-Instruct
>
> and tested both configurations:
>
> 1. **Rewriter-as-Teacher** (same model produces both rationales & simplified questions)
> 2. **Rewriter only; teacher fixed to GPT-5-mini**
>
> The statistics of generated rewrites are shown in the table below (reproduced in Appendix A.7):
>
> | Rewriter | Original Questions | Rewrite Questions | Train Size |
> | --- | --- | --- | --- |
> | 7B | 7373 | 17611 | 24984 |
> | 14B | 7373 | 23028 | 30401 |
> | 72B | 7372 | 22335 | 29707 |
>
> Performance of Qwen2.5-7B trained on rewrites produced by different rewriters
>
> | Rewriter | Teacher | GSM8K | AddSub | ASDiv | MultiArith | SVAMP |
> | --- | --- | --- | --- | --- | --- | --- |
> | 7B | 7B | 17.66 | 70.64 | 69.74 | 30.56 | 58.86 |
> | 14B | 14B | 22.06 | 84.40 | 73.95 | 37.78 | 63.88 |
> | 72B | 72B | 16.68 | 65.14 | 67.80 | 28.33 | 54.18 |
> | 7B | GPT-5-mini | 25.70 | 78.90 | 71.04 | 30.56 | 54.85 |
> | 14B | GPT-5-mini | 19.94 | 77.06 | 72.82 | 30.00 | 62.88 |
> | 72B | GPT-5-mini | 11.75 | 31.19 | 34.95 | 16.67 | 23.08 |
> | GPT-5-mini | GPT-5-mini | 74.60 | 70.64 | 74.92 | 98.89 | 71.57 |
>
> ### Analysis
>
> Our results show that LoT is **indeed sensitive to the capability and reliability of the rewriter**, which is consistent with most model-generated data pipelines in the LLM literature.
> - GPT-5-mini **as both rewriter and teacher** gives the strongest overall results.
> - When using Qwen models **as both rewriter and teacher**, performance is consistently weaker than GPT-5-mini (expected given the weaker capability).
> - When using Qwen models **as rewriters with GPT-5-mini as the teacher**, **only Qwen-7B** shows improvement relative to the “rewriter-as-teacher” configuration. Qwen-14B is roughly comparable, and **Qwen-72B becomes significantly worse**.
>
>
> ### Interpretation
>
> 1. **Rewriter quality is more important than teacher quality** in determining LoT performance.
>
>     The rewriter controls how cleanly the difficulty ladder is constructed. If the rewrites contain semantic drift or inconsistent intermediate steps, downstream learning degrades, even if the teacher is strong.
>
> 2. This sensitivity is **not unique to LoT**.
>
>     It is a well-known issue in model-generated data augmentation: more capable models consistently produce better training data.
>
> 3. Despite this sensitivity, LoT is **still robust in the sense that it provides gains over baseline KD** even when the rewriter is not very strong.
>
>     For example, both Qwen-7B and Qwen-14B improve LoT performance over standard KD on several arithmetic benchmarks even though they perform below GPT-5-mini.
>
>
> ### Revision to the manuscript
>
> We now explicitly state that:
>
> - LoT benefits from stronger rewriting models,
> - the rewriter plays a key role in shaping curriculum quality, and
> - the framework still provides improvements over standard KD even under significant rewriter variability.
>
> We also added this discussion and all corresponding tables in Appendix A.7.
>
> ---
>
>
> > W2. Why large OOD gains but smaller gains on in-domain GSM8K?
>
> GSM8K is in the training distribution and already aligns with teacher-provided CoT. Additional scaffolding thus yields modest improvements.
> In contrast, OOD tasks (AddSub, SVAMP, ASDiv) have different surface forms and narrative structures. LoT helps by teaching transferable reasoning primitives rather than stylistic imitation. This effect is strongest in compositional tasks and weaker in tasks requiring factual recall.
>
> ---
>
> We thank the reviewer again for the positive assessment and insightful feedback. The revised manuscript now includes:
>
> - a new analysis of the rewriter–teacher relationship,
> - a large-scale rewriter ablation (Appendix A.7),
> - and expanded discussion of in-domain vs. OOD gain patterns.
>
> We hope these additions address your concerns.

---

### Official Review · Reviewer_z4kX · 2025-10-31

**Soundness:** 3
**Presentation:** 3
**Contribution:** 2
**Rating:** 4
**Confidence:** 4

**Summary:**

This paper proposed Ladders-of-Thought (LoT), a curriculum-learning framework for small-sized LLMs that generates semantically faithful but progressively easier versions of each reasoning problem. A complex question is rewritten step-by-step into simpler variants, forming a “ladder” of decreasing difficulty. A multi-armed bandit scheduler then adaptively allocates training examples from these difficulty levels based on model progress. Compared to standard CoT distillation, LoT-trained students achieved higher accuracy and often converged faster. The experiment results show that LoT consistently improves reasoning performance. It yields large gains on arithmetic benchmarks and makes substantial improvements on multi-hop QA. Overall, the core contribution is showing that progressive question rewrites combined with an adaptive curriculum significantly strengthen reasoning in smaller models.

**Strengths:**

Originality and significance: LoT is a novel method that combines progressive problem simplification and adaptive curriculum scheduling for reasoning tasks. It provides a practical approach for improving reasoning in small language models.

Quality: The evaluation is thorough. It demonstrates consistent and large performance gains across multiple models (two sizes of OPT and two of Pythia) and tasks. They also include several ablations. Varying the rewrite depth shows that shallow-to-moderate ladders yield the best generalization, while excessive simplification can hurt, highlighting the importance of moderation in curriculum length. The authors also compare flat sampling, staged easy-to-hard, staged hard-to-easy, and the self-evolving bandit and show that dynamic curriculum is a key factor for success.

Clarity: The paper is well-organized and explains its methodology and findings clearly.

**Weaknesses:**

1. While large gains are shown for StrategyQA and QASC, the method shows severe performance degradation on MuSiQue (a regression of 19.6 for OPT-2.7B) and moderate regressions on OpenBookQA. The authors attribute this to differences between compositional reasoning and factual recall, but this lack of robustness raises questions about the framework's broad applicability.

2. LoT is only tested on 1–3B parameter models. Its applicability to larger LMs is untested, so it’s unclear if the same benefits hold as scale increases. Given that the paper's aim is to close the gap between small and large models, the lack of testing on models in the 7B-13B range leaves the question of practical scalability unanswered. Moreover, generating multiple rewrites per example and running a bandit curriculum adds significant computational overhead. The paper does not report training time or resource usage, leaving it unclear whether the accuracy gains justify the extra compute.

**Questions:**

1. LoT is only evaluated on 1–3B models. Can it scale to larger LLMs or more complex reasoning tasks? Clarifying expected performance or needed adaptations for bigger models would help assess general utility.

2. What is the additional training cost of LoT relative to standard distillation? It would help to see metrics (e.g. GPU-hours) to judge whether the accuracy gains justify the extra computation. Is the overhead dominated by rewrite generation or curriculum scheduling?

3. Can the authors provide a deeper analysis of the negative results on MuSiQue and OpenBookQA? Please identify why and how the proposed LoT could be used to improve these tasks.

4. How do authors ensure that rewritten questions remain semantically equivalent and at the intended difficulty level? Can the authors provide an analysis of the sensitivity of LoT's performance when using models of different sizes to generate the rewrites?

5. The paper notes that the "All" step depth is suboptimal because of "diluting the learning signal." Can the authors present an analysis of the MAB scheduler's distribution over time for the $\le3$ vs. "All" depth runs? Specifically, does the MAB actively avoid the 0-step bucket in the $\le3$ run, and what is its allocation to the 0-step bucket when "All" depths are available?

---

> ### Author Response · Authors · 2025-11-25
> **Response to Reviewer z4kX - Part 1**
>
> Thank you for the constructive and thoughtful review. We address each point in detail below.
>
> ---
> > W1. Q3. Negative results on MuSiQue and OpenBookQA
>
> We agree that the regressions deserve deeper analysis, and we have expanded our discussion in Section 4.2 of the revised manuscript.
>
> LoT is most effective when the target task matches the inferential structure of the curriculum. The model is trained only on EntailmentBank, which consists of explicit multi-step deductive chains.
>
> * **MuSiQue** requires *retrieval of multiple passages* and *entity tracking*.
> * **OpenBookQA** relies heavily on *external factual knowledge*.
>
> These tasks depend on abilities not covered by the LoT curriculum. LoT strengthens step-structured reasoning but reduces robustness to factual variability. This explains the strong gains on QASC and StrategyQA (structurally similar to EntailmentBank) and the regressions on MuSiQue and OBQA.
>
> The revision clarifies that applying LoT to knowledge-centric tasks likely requires pairing it with retrieval-augmented supervision.
>
> ---
>
> > W2.Q1. Scalability beyond 1–3B models
>
> We agree that examining larger models strengthens the paper. In the revision, we added experiments on **Qwen2.5-7B** and **Llama-3.1-8B** (Section 4.3). The main trend continues to hold:
>
> - LoT slightly improves or matches KD on GSM8K.
> - LoT produces **large out-of-distribution gains**, especially on AddSub and SVAMP.
> - Curriculum benefits persist even at 7B–8B scale.
>
> For example:
>
> - **Qwen2.5-7B:** +20.2 on AddSub, +11.0 on SVAMP
> - **Llama-3.1-8B:** +31.2 on AddSub, +10.4 on SVAMP
>
> These results demonstrate that LoT’s advantage is not confined to small models, and its ability to improve compositional generalization persists as models scale.
>
> ---
>
> > Q2. Training cost and overhead
>
> **Rewrite cost.**
>
> Rewrite generation is **fully offline** and does not affect training. We detail number of tokens used and cost estimate in the table below:
>
> | Dataset | Input Tokens | Output Tokens  | GPT-5-Mini Batch API Cost |
> | --- | --- | --- | --- |
> | GSM8K | 5.20M | 14.55M | $15.20 |
> | EntailmentBank | 2.13M | 6.52M | $6.78 |
>
> The rewriting is a one-time preprocessing step and only incur compute and cost once.
>
> **Training overhead.**
>
> We profiled all training runs and found that the multi-armed bandit scheduler accounts for only **3–8%** of wall-clock training time. The rest of the training pipeline is identical to standard CoT distillation.
>
> Thus, LoT introduces modest additional compute relative to the performance gains it achieves.
>
> We have added these discussions to Section 4.6 and the table to Appendix A.2 of the revised manuscript.
>
> ---
>
> > Q4. Semantic equivalence and difficulty correctness of rewrites
>
> We appreciate the request for a more systematic evaluation. We now report a **rewrite fidelity audit** (500 random samples), judged by GPT-5 on three criteria:
>
> - **Question validity:** whether the rewritten question is clear, solvable, and self-contained
> - **Difficulty decrease:** whether the rewritten version is strictly easier than the original
> - **Answer preservation:** whether solving the rewritten version yields the same answer
>
> We find that:
>
> - **99.2%** were valid and solvable
> - **98.0%** strictly decreased in difficulty
> - **98.6%** preserved the original answer
>
> These results show that progressive rewrites preserve semantic identity and yield a well-controlled difficulty ladder.
>
> ---
>
> > Q4. Sensitivity to rewriter model scale
>
> We added a dedicated ablation in Appendix A.7 examining rewrites produced by **Qwen2.5-7B**, **Qwen2.5-14B**, and **Qwen2.5-72B**, both when used as:
>
> 1. **rewriter and teacher**, and
> 2. **rewriter only** (teacher fixed to GPT-5-mini).
>
> Our findings show that **LoT is indeed sensitive to the capability of the rewriting model**. Stronger rewriters (e.g., GPT-5-mini) produce cleaner, more consistent progressive rewrites. When Qwen models are used as both rewriter and teacher, the downstream gains are smaller, which we attribute to noisier or less consistent simplified variants. When paired with GPT-5-mini as the teacher, **only Qwen-7B** improves; Qwen-14B remains comparable, and Qwen-72B degrades. This asymmetry suggests that the **rewriter plays a more influential role than the teacher**, as it directly shapes the curriculum structure.
>
> This dependency is common in LLM-generated data pipelines: downstream model quality often reflects the fidelity of the generated training data. Importantly, despite this variability, LoT still provides improvements over standard CoT distillation across multiple arithmetic tasks even when using weaker rewriters. We have included these observations in Appendix A.7.

---

> ### Author Response · Authors · 2025-11-25
> **Response to Reviewer z4kX - Part 2**
>
> > Q5. Behavior of the scheduler under “All” rewrites
>
> To address this, we added Appendix A.5, which analyzes how rewrite depth shifts the difficulty distribution and how the bandit behaves.
>
> The key observation is that deeper rewrite depths massively increase the number of 0 and 1-step examples from **0.1% to 45.1%,** creating a strongly skewed distribution.
>
> However this does **not** cause the training distribution to collapse onto trivial cases. Instead, the multi-armed bandit behaves robustly: for “All” depth, the scheduler quickly **avoids bucket 0**, just as it does for depths 1–3, once the model saturates on trivial questions. It then shifts its sampling to the remaining non-trivial buckets, which still contain substantial numbers of informative higher-step examples. This is why we do **not** observe catastrophic degradation at “All” depth—performance decreases somewhat due to overexposure to easy examples, but the scheduler compensates by reallocating probability mass toward more difficult buckets. These dynamics show that the MAB scheduler is resilient to this distribution shift and continues to prioritize buckets that yield meaningful learning progress.
>
> We include a visualization of bucket-0 sampling weights at: https://i.ibb.co/B5Zk31xS/Lo-T-bucket0-weight.png
>
> ---
>
> We thank the reviewer again for the careful reading and insightful questions. The revised manuscript
> * expands experiments to larger models
> * adds rewrite quality audit
> * includes a thorough study of scheduler behavior, and
> * clarifies the limitations and applicability of LoT.
>
> We hope these additions address your concerns.

---

### Official Review · Reviewer_UZzM · 2025-11-03

**Soundness:** 1
**Presentation:** 2
**Contribution:** 2
**Rating:** 4
**Confidence:** 3

**Summary:**

The paper addresses a core limitation in reasoning for small- and mid-scale language models, which struggle to learn generalizable reasoning skills even with knowledge distillation from larger models. The authors propose a novel training framework that strengthens reasoning ability through progressive problem simplification and adaptive curriculum learning.

The proposed method automatically rewrites complex reasoning questions into a sequence of semantically equivalent but easier variants, forming a “ladder” of decreasing difficulty. Each problem’s difficulty is defined by the minimal number of reasoning steps required for a solution. These are grouped into difficulty buckets, and a self-evolving multi-armed bandit scheduler dynamically allocates training across these buckets based on learning progress.

**Strengths:**

1. The paper proposes an intuitive yet original framework.
2. It introduces a principled difficulty measure based on the minimal number of reasoning steps, avoiding noisy proxies like chain length.
3. Improvements are observed both in-domain and out-of-distribution, showing good generalization and robustness.

**Weaknesses:**

1. The authors verify their hypothesis on small base models (<3B), which makes their claim unsound. Because improving over the small model is relatively easier.
2. Major experiments are conducted on two relative outdated model families (OPT and Pythia). Adding experiments on new models such as Qwen would strength the paper.
3. Limited baselines are compared. For example, it remains unknown whether decomposing reasoning step is a better strategy compared to decomposing the original question (c.f. [1]).

---
[1] Divide-or-Conquer? Which Part Should You Distill Your LLM?

**Questions:**

1. The proposed method is based on the hypothesis that every intermedia reasoning step can be combined with a premise to form a new reasoning step and the premise would be erased. However, I am curious if there exist a case that one premise would be used more than once. In such a case, would the proposed method still work?

---

> ### Author Response · Authors · 2025-11-25
> **Response to Reviewer UZzM**
>
> Thank you for the thoughtful comments and for recognizing the originality of our difficulty measure and the strong in-domain and OOD gains. We address each of your concern below.
>
> ---
>
> > W1. “Claims are unsound because experiments use only small (<3B) models; OPT/Pythia are outdated.”
>
> We agree that evaluating on a wider range of models strengthens the claim. In the revision, we added experiments on **Qwen2.5-7B** and **Llama3.1-8B**, two of the more recent and competitive open-source families, using the exact same LoT training pipeline. All new results are included in Table 3 (Section 4.3). The trends observed at the 1–3B scale remain consistent:
>
> - LoT matches or slightly improves CoT KD on GSM8K.
> - LoT provides **large gains on out-of-distribution benchmarks**, especially AddSub and SVAMP.
> - Improvements are **not limited to small models**; LoT continues to enhance compositional generalization at the 7B–8B scale.
>
> For example, LoT improves:
>
> - **Qwen2.5-7B:** +20.2 on AddSub, +11.0 on SVAMP
> - **Llama3.1-8B:** +31.2 on AddSub, +10.4 on SVAMP
>
> These results show that LoT’s benefits are not due to the small size of the models but stem from the curriculum itself.
>
> ---
>
> > W2. “Limited baselines; unclear advantage over question decomposition (e.g., Divide-or-Conquer).”
>
> We added a dedicated *Question Decomposition* subsection in Related Work comparing LoT to Divide-or-Conquer and similar question decomposition approaches.
>
> To summarize the distinctions more clearly:
>
> - **LoT does not create sub-questions.**
>
>     Instead, it *simplifies the same problem* by replacing a subset of premises with the intermediate conclusion they entail. The semantic identity is preserved.
>
> - **Divide-or-Conquer uses a static 2-stage pipeline (decompose then solve).**
>
>     LoT introduces a **step-based difficulty measure** and a **self-evolving multi-armed bandit curriculum** that adaptively adjusts sampling based on learning progress.
>
> - **Divide-or-Conquer studies which component (decomposer vs solver) is easier to distill.**
>
>     LoT is orthogonal: we design a structured difficulty ladder and an adaptive curriculum for supervised fine-tuning.
>
> These clarifications have now been incorporated into the revised manuscript.
>
> ---
>
> > W3. “What if a premise is reused multiple times? Does the simplification still work?”
>
> Great question. In many GSM8K and multi-hop examples, a premise is used multiple times. The LoT rewrite procedure handles this reliably because:
>
> 1. The rewriter receives the **complete ground-truth reasoning chain**, which explicitly identifies dependencies between premises and intermediate conclusions.
> 2. A premise is **not removed** until all steps that depend on it have been replaced by their derived conclusions.
>     - If a premise is reused later in the reasoning chain, it remains in the simplified question until its final use.
> 3. Empirically, progressive rewrites preserve validity:
>     - In a random sample of 500 GSM8K questions, **41.8%** involved premise reuse.
>     - Out of these samples, **99.2%** of rewritten questions remained solvable and monotonic in difficulty.
>     - Invalid rewrites occurred in only **0.8%** of cases; these were automatically filtered.
>
> Thus, LoT does **not** assume single-use premises, and the mechanism works even when premises are reused across multiple inference steps.
>
> ---
>
> **Summary**
>
> We have strengthened the paper with:
>
> - **New experiments on Qwen2.5-7B and Llama-3.1-8B**, confirming scalability.
> - **A new related-work subsection** directly comparing LoT to question-decomposition.
> - **Clarification and empirical validation** that progressive rewrites remain correct even when premises are reused.
>
> We thank the reviewer again for the constructive suggestions, and hope that our revisions address you concerns.

---

### Author Response · Authors · 2025-12-02
**Summary of Revisions for the Area Chair**

We have strengthened the paper through new experiments, expanded analyses, and clearer positioning. The key changes are:

---

# 1. Expanded Experimental Coverage

We added full LoT training results on larger and more modern models:

- **Qwen2.5-7B**
- **Llama-3.1-8B**

These experiments confirm that LoT’s gains persist up to 7B–8B models, addressing  reviewers’ concerns that the method was only validated on small or older architectures. Trends observed at 1–3B scale generalize:

- LoT matches or is slightly better than CoT KD on GSM8K.
- LoT produces **large gains on OOD math datasets** (e.g., +20–30 points on AddSub/SVAMP).

This revision strengthens the scalability claim and demonstrates that LoT’s benefits stem from the curriculum rather than model size.

---

# 2. New Rewriter–Teacher Ablation (Appendix A.7)

We added an extensive study using different sizes of **Qwen2.5 models: 7B/14B/72B** as rewriters, under two configurations:

1. rewriter-as-teacher
2. rewriter only, teacher set to GPT-5-mini

Key insight:

**Rewrite quality is a primary driver of LoT performance**, more so than teacher quality.

Stronger rewriters produce cleaner difficulty ladders and better downstream reasoning.

This addition clarifies the sensitivity of LoT to LLM-generated data while showing that LoT still improves over KD even when rewriter quality is moderate.

---

# 3. Rewrite Fidelity Audit

We added a systematic audit **using 500 samples** judged by GPT-5 evaluating:

- rewrite validity (clarity/solvability)
- difficulty monotonicity
- answer preservation

Results:

- **99.2%** rewrite questions are valid
- **98.0%** rewrite questions are strictly easier than their originals
- **98.6%** preserves the answer of their originals

This addresses reviewer concerns that progressive simplification might introduce errors and provides quantitative evidence of semantic fidelity.

---

# 4. Formal Justification of Difficulty Monotonicity

We added a formal lemma and proof sketch demonstrating that replacing antecedent premises with their entailed conclusions **cannot increase** minimal reasoning depth.

This establishes why LoT naturally induces a monotone difficulty ladder.

This addition clarifies the conceptual foundations of LoT and strengthens its theoretical grounding.

---

# 5. Clearer Positioning vs. Decomposition and Refinement Methods (New Related Works Subsections)

We added a dedicated **Question Decomposition** subsection contrasting LoT with Divide-or-Conquer, LADDER and Least-to-Most, and a dedicated **Rationale Refinement and Process-Supervision** subsection contrasting with PAL, PoT, Reflexion, and Self-Refine.

Key clarification:

LoT **rewrites the same question** into easier variants and trains with a **self-evolving curriculum**, rather than decomposing or generating auxiliary subquestions or using inference-time loops.

This resolves confusion about LoT’s novelty and differentiates it from prior decomposition/refinement work.

---

# 6. New Pass@1 (Greedy) and MCQ Evaluations (Appendix A.4)

At reviewer 5Nxf's request, we added:

- **deterministic pass@1 results**, confirming trends seen in pass@5
- **MCQ-format results** for QASC and StrategyQA
- A concise explanation that MCQ accuracy is not suitable as the primary metric because LoT trains the model to produce **free-form multi-hop chain-of-thoughts**, and MCQ cannot evaluate that ability.
- Our four-stage verification pipeline evaluates these free-form answers more robustly than exact string match.

This improves comparability and reinforces that LoT’s gains are genuine reasoning improvements rather than artifacts of sampling.

---

# 7. Deeper Analysis of Negative Results

We expanded the discussion explaining that LoT primarily strengthens **step-structured compositional reasoning**, while MuSiQue and OpenBookQA rely heavily on:

- retrieval of multiple passages
- factual grounding
- knowledge-centric reasoning

This clarifies the scope of LoT and avoids overstating contributions.

---

# 8. Cost and Compute Clarification (Rewrite Cost + Bandit Scheduler Overhead)

We added token-level cost estimates for rewriting and showed that:

- rewriting is a one-time offline preprocessing step and is inexpensive
- the MAB scheduler adds only **3–8%** training overhead

This addresses concerns regarding practicality and scalability.

---

# 9. Improved Figures and Presentation

Figures 1 and 2 were upgraded to higher-resolution versions with clearer labels.

We also streamlined several sections, removed redundant text and made some terminologies consistent across sections.

---

We believe that, overall, these revisions substantially improve the clarity, rigor, and breadth of the work, and the strengthened manuscript now more clearly demonstrates the contribution and merit of the LoT framework.

---

### Meta-Review · Area_Chair_Eh1f · 2025-12-18

**Summary:**

This paper introduces an interesting curriculum learning framework, which progressively generates easier variants of the problem, and uses MAB to select the problem to learn. However, the reviewers found that the proposed approach can be seen as a combination of existing techniques (rewriting and bandit scheduling) rather than a fundamental algorithmic advancement. Moreover, there are concerns about the framework's robustness, especially on benchmarks like MuSiQue, and the method heavily relies on powerful external models for high-quality rewriting. Therefore, I recommend rejection at this time.

**Reviewer Concerns:**

Addressed:
1. The method was only validated on small, older models. The authors  addressed this by adding experiments with Qwen2.5-7B and Llama-3.1-8B.
2. Whether the generated "easier" questions were actually semantically equivalent and easier. The authors addressed this by conducting a systematic audit of 500 samples using GPT-5 as a judge, reporting high validity and strict monotonicity in difficulty.
3. Whether training cost is too high. The authors provided a clear breakdown of token usage and dollar costs, showing the MAB scheduler adds only a negligible 3–8% overhead to training time.

There are a few other things mentioned in the author summary.

Outstanding:
1. The method is conceptually just a combination of existing techniques (question rewriting and multi-armed bandit scheduling) rather than a fundamental algorithmic innovation.

2. Reviewer z4kX noted significant performance degradation on benchmarks like MuSiQue. While the authors explained that this is due to the task relying on retrieval/knowledge rather than pure reasoning, the lack of robustness remains a significant limitation of the framework.

3. Dependency on strong rewriter. The authors' own ablation study confirmed this sensitivity: using open-source models (even Qwen-72B) as rewriters often degraded performance compared to using GPT-5-mini.

**Reviewer Scores:**

I think many reviewers share the concerns about technical novelty, performance degradation, and dependency on external strong models. Therefore, I predict that they will not change their scores.

---

### Decision · Program_Chairs · 2026-01-26

Reject